**Subject Category:**
Biology (whole organism)

ecology

climate change, drought, physiological stress, survival, resource availability

**Author for correspondence:**
Christopher Young
e-mail: christopher.young@uleth.ca

# Climate induced stress and mortality in vervet monkeys

Christopher Young[1,2,3], Tyler R. Bonnell[3], Leslie R. Brown[2], Marcus J. Dostie[3,4], Andre Ganswindt[1], Stefan Kienzle[2,4], Richard McFarland[5,6], S. Peter Henzi[2,3] and Louise Barrett[2,3]

[1]Endocrine Research Laboratory, Mammal Research Institute, Faculty of Natural and Agricultural Science, University of Pretoria, Pretoria, Republic of South Africa
[2]Applied Behavioural Ecology and Ecosystems Research Unit, University of South Africa, Pretoria, Republic of South Africa
[3]Department of Psychology, and [4]Department of Geography, University of Lethbridge, Lethbridge, Alberta, Canada
[5]Department of Anthropology, University of Wisconsin-Madison, Madison, WI, USA
[6]Brain Function Research Group, School of Physiology, University of the Witwatersrand, Johannesburg-Braamfontein, Republic of South Africa

CY, 0000-0001-8919-2093; TRB, 0000-0001-6041-5177;
RM, 0000-0001-8245-9269; SPH, 0000-0001-6175-1674

As the effects of global climate change become more apparent, animal species will become increasingly affected by extreme climate and its effect on the environment. There is a pressing need to understand animal physiological and behavioural responses to climatic stressors. We used the reactive scope model as a framework to investigate the influence of drought conditions on vervet monkey (*Chlorocebus pygerythrus*) behaviour, physiological stress and survival across 2.5 years in South Africa. Data were collected on climatic, environmental and behavioural variables and physiological stress via faecal glucocorticoid metabolites (fGCMs). There was a meaningful interaction between water availability and resource abundance: when food availability was high but standing water was unavailable, fGCM concentrations were higher compared to when food was abundant and water was available. Vervet monkeys adapted their behaviour during a drought period by spending a greater proportion of time resting at the expense of feeding, moving and social behaviour. As food availability decreased, vervet mortality increased. Peak mortality occurred when food availability was at its lowest and there was no standing water. A survival analysis revealed that higher fGCM concentrations were associated with an increased probability of mortality. Our results suggest that with continued climate change, the increasing prevalence of drought will negatively affect vervet abundance and distribution in our population. Our study

contributes to knowledge of the limits and scope of behavioural and physiological plasticity among vervet monkeys in the face of rapid environmental change.

## 1. Introduction

Climate change is not only exposing animals to progressively warmer climates [1,2], but is also imposing a higher frequency of extreme weather events, including heat waves, cold snaps, storms, flooding and drought [1,2]. Such instability and unpredictability in environmental conditions is widely recognized as a major source of stress to animals [3], and extreme conditions have been demonstrated to negatively affect individual fitness and population survival (e.g. [4–6]). As the effects of global climate change become increasingly more apparent, a greater number of species and environments will be affected by extremes of both high and low temperatures [2], coupled with the knock-on effects that climate-induced events have on the availability of suitable habitat, shelter, and food and water resources. Therefore, there is a pressing need to understand the scope and limits of animal stress responses to local climatic extremes, and not just their response to average increases in global temperatures (see [7]).

Environmental temperature has been shown to have a direct impact on an animal's stress physiology, with extremes of both heat and cold being linked to elevated cortisol levels in a range of animal taxa (heat stress: [8–10]; cold stress: [11–13]). High ambient temperatures, especially in combination with low rainfall, can also drive reductions in food availability and hence food intake, which can further increase physiological stress ([11]; reviewed by [3]). Any effect of climatic factors on cortisol levels will therefore be compounded by attendant variation in resource availability, and perhaps also by increased levels of feeding competition [14]. Under extreme conditions, resources may become completely depleted and have equally extreme effects on individual health and well-being [2]. For example, drought conditions (a water shortage caused by a period of abnormally low rainfall) led to high mortality of aardvarks (*Orycteropus afer*), most likely due to resource depletion and starvation [5].

The reactive scope model [15] offers an excellent framework in which to situate the study of stress in relation to changes in ecological conditions. In brief, the reactive scope model conceives of the stress response—the well-orchestrated physiological changes that occur when cortisol levels are elevated—as the animal's attempt to restore or maintain homeostasis when exposed to environmental variability or changes in their physiological status. Such alterations can be predictable (e.g. regular daily or seasonal shifts in daylight, temperature or rainfall, and life-history changes, such as the onset of reproduction), or unpredictable (e.g. floods, droughts and physical injury). Accordingly, the 'reactive scope' of a healthy individual can be divided into two components: the response to circadian or seasonal is known as predictive homeostasis, while increases of levels of a physiological mediator above the normal circadian or seasonal range (i.e. increases outside the range of predictive homeostasis) is known as reactive homeostasis. The reactive scope of an individual thus defines the physiological constraints within which a healthy animal operates.

Maintaining a mediator within the reactive homeostasis range incurs costs in terms of both direct energy consumption and opportunity costs (i.e. the ability to maintain other tissue function), and these costs increase the longer the mediator remains in the reactive homeostasis range. The accumulation of these costs leads to 'wear and tear' (also known as allostatic load; [15]). If physiological mediators exceed the range of the normal reactive scope, the animal is pushed into 'homeostatic overload': here, levels of the physiological mediator become so high that they themselves begin to cause physiological disruption [15,16].

Accumulation of wear and tear over time can result in a decline in animal's homeostatic overload threshold, so that it enters the overload phase sooner. Such wear and tear often can be repaired if the stressor is subsequently removed. In some cases, however, the homeostatic overload threshold is permanently lowered, such that animals experience an overall reduction in their reactive scope. The reactive scope model thus shows how, as a consequence of variation in individual experience, animals may vary in their ability to maintain homeostasis in the face of ecological stressors. Thus, the same stressor may exert a differential impact within a given population or group of animals depending on the degree of variation in their respective reactive scopes [15,16].

Here, we take advantage of a period of drought in the Karoo region of South Africa, to investigate its possible impact on vervet monkey (*Chlorocebus pygerythrus*) physiological stress levels (measured here as faecal glucocorticoid metabolites; fGCM concentrations), activity and survival probabilities, using the

reactive scope model as a guiding framework (that is, we assume that levels of faecal glucocorticoids provide an indication of how hard the animal is working to restore homeostasis). The Karoo is a temperate region, normally characterized by hot, dry summers and cold, wet winters. Temperatures can reach highs of 45°C in the summer, and lows of −10°C in the winter, with average rainfall in the region of approximately 450 mm yr$^{-1}$ [17]. During 2016 and 2017, however, the region experienced very low levels of rainfall (2016 = 220.8 mm and 2017 = 227.4 mm) resulting in reduced food and, most notably, the complete absence of standing water for extended periods (i.e. no drinking water was available in the monkeys' home ranges).

Variation in fGCM concentrations during non-drought conditions should reflect predictive homeostasis in response to seasonal variation in temperature and food availability. We hypothesize that drought conditions, as an acute environmental stressor, should trigger reactive homeostasis and lead to higher fGCM concentrations than those seen during non-drought conditions. Specifically, we predict an interaction between water availability and both daily temperature and food availability, with elevated fGCM concentrations associated with the loss of standing water over and above the direct effects of temperature and food availability [18]. We further predict that there will be individual differences in fGCM concentrations and hence reactive scope, reflecting variation in previous ecological experience and thus variable wear and tear.

In addition, we explore potential behavioural trade-offs during drought conditions, to determine whether and how the behaviour of individual vervets differs during drought conditions compared to those within the typical range. We have previously shown that, during non-drought conditions, animals in our population rest more during elevated temperatures and feed more at lower temperatures [19]. Therefore, we predict that individuals will rest more to compensate for heat stress and lack of standing water, particularly when temperatures are high. Finally, we investigate the nature of the relationship between fGCM concentrations and mortality, food and water availability. We predict that a decline in these ecological resources will be associated with higher mortality, and that elevated fGCM concentrations will differentiate non-survivors from survivors.

# 2. Methods

## 2.1. Study site and behavioural data collection

Behavioural and hormonal data were collected from three habituated groups of vervet monkeys (RST, RBM and PT) in the Samara Private Game Reserve, South Africa. All individuals are individually recognizable via distinct facial and body markings. Groups are followed for 10 h per day, 5 days per week [20,21]. During the study period for these analyses (April 2015–August 2017), our three groups showed variation in group size: RST = 47–65 ($N_{male}$ = 5–10; $N_{female}$ = 8–15), PT = 33–43 ($N_{male}$ = 3–9; $N_{female}$ = 8–10), RBM = 49–62 ($N_{male}$ = 6–13; $N_{female}$ = 8–18). Behavioural data were collected from all adult group members during 10 min scan samples [22] conducted every 30 min throughout the day. Behaviours were divided into four mutually exclusive categories: moving, foraging, resting and social behaviour (i.e. grooming and playing).

## 2.2. Dominance rank

Dominance rank was calculated using standardized David's scores [23]. We collected decided agonistic interactions between all adult individuals in our groups on an ad libitum basis ($N_{RST}$ = 5518; $N_{RBM}$ = 3860; $N_{PT}$ = 4530). The loser of an interaction was the individual who was last to show submission. These agonistic interactions were divided into three-month windows in order to generate reliable winner/loser matrices [24]. We have shown previously that male and female vervet monkey hierarchies are interdigitated in our population [24]. As a result, we used a single combined hierarchy that encompasses both males and females. We used the DomiCalc macro in Excel in order to generate David's scores for each three-month period and each group separately [25]. We used standardized David's scores to facilitate comparisons between groups and time periods [24,26]. For each three-month time period and each group, separately, the David's score for each individual was divided by the highest score in that time period, to give the highest-ranking individual a score of 1. We then time matched the date of faecal sample collection to the individual's David's score from that three-month time period to give their standardized dominance rank for the time of each sample collected.

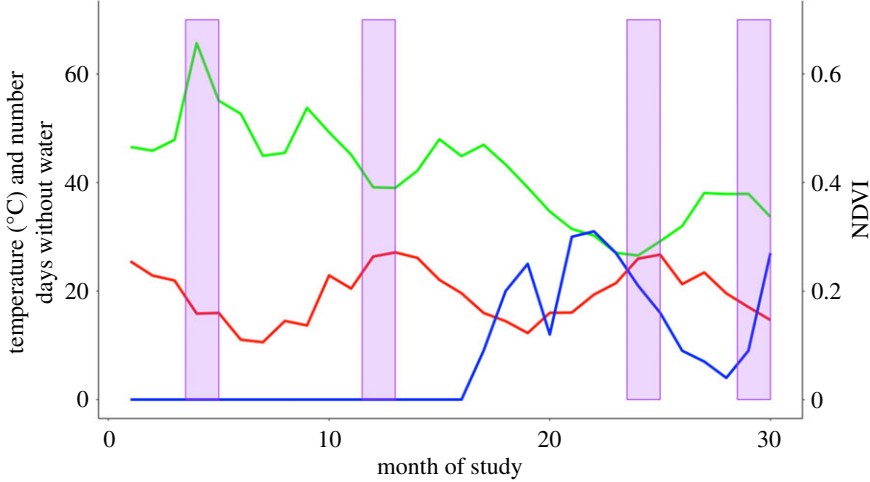

**Figure 1.** Schematic graph to illustrate the overall relationship between food availability (monthly average normalized difference vegetation index (NDVI), green line), mean monthly daily average temperature (red line) and the number of days without water in the previous 30 days (blue line) during the entire data collection period. On the x-axis is the month of study from 1 = January 2015 to 30 = June 2017. The y-axis (left) shows temperature in degrees Celsius and the number of days without water (both on the same scale). The y-axis (right) shows the NDVI score on a scale of 0–1 with 0 being the lowest score and 1 being the highest. The purple blocks indicate the data collection periods (from left to right): 'cold-wet' (April–June 2015), 'hot-wet' (December 2015–February 2016), 'hot-dry' (December 2016–February 2017) and 'cold-dry' (April–June 2017). These are also the periods when faecal samples were collected.

## 2.3. Mortality

We recorded mortality of all juvenile and adult female vervets. We excluded adult males from this analysis as it was not possible to determine whether disappearance from the group was due to mortality or emigration (males migrate several times in their lifetime and may do so throughout the year: [27,28]). If an infant, juvenile or adult female disappeared and their body could not be found, the date of death was given as the last date on which the individual was observed in the group.

## 2.4. Measuring fGCM concentrations

We examined fGCM concentrations across the three groups of vervet monkeys during four distinct climatic conditions, across a period of six weeks for each condition. These were 'cold-wet' (April–June 2015), 'hot-wet' (December 2015–February 2016), 'hot-dry' (December 2016–February 2017) and 'cold-dry' (April–June 2017; see figure 1). This gave us two hormone samples per animal per period. Due to migration events, deaths or less than two samples being collected in a time period, not all individuals were present in all periods. There were a total of 59 individual animals sampled for fGCM concentrations: cold-wet = 64 ($N = 32$ individuals), hot-wet = 104 ($N = 52$), hot-dry = 96 ($N = 48$) and cold-dry = 82 ($N = 42$).

Faecal samples were collected ad libitum twice per month from all adults within our three groups. Samples were collected following the protocol of Young *et al.* [21]: each sample was collected from a positively identified individual after defecation and within 15 min of defecation. Samples were collected from where they fell after defecation generally on leaf litter or the ground and a 2–5 g piece of faecal material was transferred into a plastic vial following physical homogenization of the entire faecal sample using a wooden stick. Faecal samples were checked to ensure there was no contamination with urine either during excretion or on the substrate where the sample landed. Vials were stored on ice in a thermos in the field before transfer to a −20°C freezer at the end of the day. Samples were stored until transport on dry ice to the Endocrine Research Laboratory, University of Pretoria, for analysis.

We have previously confirmed the most appropriate enzyme immunoassay (cortisol assay) for vervet monkey faecal sample analysis of fGCM concentrations on the study population [21]. The full extraction process is reported in Young *et al.* [21]. In brief, samples were lyophilized and then pulverized and sieved to remove seeds and fibrous matter. The resulting fine faecal power was weighed and 0.10 g was extracted. The extraction process involved 15 min of vortexing with 80% ethanol in water (3 ml)

**Table 1.** fGCM concentrations for the four six-week data collection time periods. Shown are the mean values ± standard deviations of fGCM concentrations across all individuals for each period plus the range of values (i.e. the minimum and maximum values; $N = 346$). Also included are the average demonstrated reactive scope (DRS) and the coefficient of variance of the DRS (DRS$_{CV}$) across all individuals in each study period. All fGCM values are nanograms per gram dry weight (ng g$^{-1}$).

| date | time period | fGCM mean ± s.d. | fGCM range | DRS | DRS$_{CV}$ |
|------|-------------|------------------|------------|-----|-----------|
| Apr–May 2015 | cold-wet | 50.70 ± 30.84 | 16.04–172.12 | 310.74 | 60.86 |
| Jan–Feb 2016 | hot-wet | 57.38 ± 57.28 | 14.71–446.94 | 1338.81 | 99.83 |
| Jan–Feb 2017 | hot-dry | 77.91 ± 73.32 | 21.76–578.47 | 821.81 | 94.11 |
| May–June 2017 | cold-dry | 65.97 ± 50.24 | 25.97–414.80 | 709.15 | 76.16 |

followed by 10 min of 1500$g$ centrifugation. Then, 1.5 ml of the resultant supernatants were transferred into microcentrifuge tubes for hormone analysis. For the hormone analysis we followed standard procedures of the Endocrine Research Laboratory, University of Pretoria, as described previously [29], using the cortisol enzyme immunoassay (EIA) [21]. Inter- and intra-assay coefficients of variation of high- and low-value quality controls were: 4.64–5.96 and 8.13–11.60% respectively. All steroid concentrations are given as ng g$^{-1}$ faecal dry weight.

Following MacLarnon *et al.* [30], we additionally calculated the demonstrated reactive scope (DRS) and coefficients of variation of DRS (DRS$_{CV}$) for each time period across all individuals. To do so, we calculated the mean fGCM concentrations and the standard deviation of fGCM concentrations for each time period as well as the maximum and minimum values. To calculate the DRS, we subtracted the minimum score from the maximum, and divided by the minimum and multiplied by 100. For the DRS$_{CV}$ we divided the standard deviation by the mean fGCM concentrations per time period and multiplied by 100 (table 1). We include these values as an overview of the differences between time periods, but do not use the DRS in our statistical analyses as the latter are conducted at the individual level.

## 2.5. Climatic variables

Our climatic and environmental variables were extracted in a manner that made them ecologically or physiologically relevant (within the constraints of our sampling regime and resolution). For each faecal sample analysed, we calculated mean daily globe temperature for 2 days prior to collection of the sample. We selected a 2-day window for temperature, as this represents the time lag for steroid excretion in the faeces in this population [21]. Globe temperature is a measure including direct sun radiation, air temperature and air velocity in order to resemble the thermal conditions experienced by an organism [31]. A 150 mm diameter black globe thermometer was used to record environmental temperature every 30 min from a centralized weather station at the field site (Hobo U30-NRC, Onset Computer Corporation, USA).

In addition to temperature, we recorded the availability of water within each group's home range. Water is available from natural sources only. The Melk River, which runs through the middle of the study site, is the main source of water [17], along with temporary water catchment on firm ground and rocks during rain showers. Water availability in the home range of each group was recorded separately on a daily basis as either present or absent. We recorded each day if the river was flowing, if there were pools of water available in the home-range of each group or if water was completely absent. If pools were present the monkeys were observed drinking from the pools each day until the pools dried up. For each faecal sample analysed, we calculated the number of days during the previous 30 days that water was unavailable within the home-range of the individual, hereafter 'days without water'. We used 30 days for water availability, as we were trying to capture the cumulative effect of a chronic shortage of water. Using a shorter window (such as the 2 days for the temperature variable) would not capture chronic shortage, and potentially would generate a false impression of how much water was present in the home range. That is, during the drought period, the river dried up completely, with the result that there was no access to water for the monkeys for several days and even weeks at a time. When it did rain, high temperatures (in excess of 45°C) meant that any small pools of water dried up rapidly. So, if a faecal sample was collected in the 1 or 2 days following a rain shower, but this was actually the only rainfall for the whole month and water was present for

only a few days, it would give a misleading impression of how much water was available to the monkeys overall.

## 2.6. Food availability

Food availability was measured by calculating the average normalized difference vegetation index (NDVI) from all pixels in the study area using the moderate resolution imaging spectroradiometer MOD13Q1 16-day vegetation indices at a 250 m resolution [32]. NDVI has been used in many ecological studies as a proxy measure of biomass [33,34]. NDVI measures the amount of biomass or chlorophyll activity in a given area by calculating the difference between the visible red and near infrared bands divided by their sum [35]. This measure provides a range of values between −1 and 1, where negative values indicate an absence of vegetation, positive values approaching 0 indicate less green vegetation and positive values approaching 1 indicate larger concentrations of green vegetation [33]. Therefore, more photosynthetically active areas have a higher NDVI score and are considered to be more abundant in biomass [36,37]. We have previously shown that group movements and home range size are predicted by NDVI variation across the same period ([38]; see also [39]), and we use this measure as a proxy for food availability in the area. Images were downloaded twice per month. We looked at the impact of the current food availability on fGCM concentrations by looking at the NDVI value relating to the closest image prior to faecal sample collection.

## 2.7. Statistical analysis

To test our predictions, we used a Bayesian multilevel statistical approach. We also explored variation in the structure of the random effects to examine the influence of these factors on an individual level.

We used the function 'brm' from the R package 'brms' [40] implemented in R using 'RStan' [41] to estimate environmental influences on fGCM concentrations. Models were fitted with Hamilton Markov chains and run in R v.15.6.0 [42]. This approach allowed us to examine the main effects of predictor variables on the response variable, along with the influence of variance within the random effects, and the influence of individual variance on our response variable [43]. We present summary statistics for posterior means, standard errors (s.e.) and 95% credible intervals (CI) for the main effects, and individual variance within the random effects. All continuous variables were standardized by subtracting the mean and dividing by two times the standard deviation to facilitate comparisons of the effect sizes across continuous and dichotomous variables [44]. We used weakly informative priors (mean = 0 and s.d. = 10 for all continuous variables), four chains and 4000 iterations [40,45]; this provided a large enough sampling pool to allow for posterior sampling and model convergence. All $\hat{r}$ were less than 1.1, which indicates that our models converged, while whole posterior predictive checking indicated that no model assumptions were violated [46]. We used the 'bayes_R2' function to generate marginal $R^2$, conditional $R^2$ and conditional $R^2$ minus random slopes but still including random intercepts [47].

## 2.8. Model 1: Influence of environmental conditions on fGCM concentrations

In order to examine the influence of environmental conditions on fGCM concentrations we fitted our data using a linear mixed model with a lognormal error structure. Our response variable was individual fGCM concentrations and our predictor variables were (1) sex (male or female), (2) food availability (calculated from NDVI), (3) water availability (number of days without water in the previous 30 days), and (4) mean daily average temperature of the previous 2 days (model 1, $N = 346$). We also included two interaction terms between (i) temperature and water availability and (ii) food availability and water availability. We included standardized dominance rank as a control factor. We specified individual identity (ID) to control for multiple samples, and included random slopes and random intercepts for all of our predictor variables [48].

As climatic variables can show multicollinearity, we first ran general linear models without the random effects and examined the variance inflation factors (VIFs) using the R package 'car' and the function 'vif' [49]. We found that for model 1, VIFs were greater than 4 and the food availability and temperature variables showed multicollinearity. We therefore ran two separate models: (1) Model1$_{temp+water}$ (with the interaction term of temperature and days without water but no food availability), (2) Model1$_{food+water}$ (with the interaction term of food availability and days without water but no temperature variable). In doing so the VIFs reduced to less than 4 for both models. We

then applied the widely applicable information criterion (WAIC: [50]) (the generalized version of the more familiar AIC) to determine which of our models gave the best fit to our data, where the lowest WAIC score indicates the best fitting model. We found that the $Model1_{food+water}$ showed lower WAIC, 2718.9 (±44.7 s.e.) than $Model1_{temp+water}$ (2728.2 ± 44.6 s.e.). We therefore present the results of $Model1_{food+water}$ (for the results of $Model1_{temp+water}$ see electronic supplementary material, table S1).

## 2.9. Model 2: Trade-off of behaviours under different environmental conditions

In order to examine whether drought conditions were associated with any change in behaviour, we built a multilevel multinomial behaviour model [51]. These models estimate the likelihood of a given behaviour from a set of categorical behaviours occurring at any given time in relation to a reference behaviour, while controlling for pseudo-replication (i.e. many data points from the same individual). In our model ($model2_{behaviour}$), we set the categorical variable of behaviour (social, feeding, resting and moving) as our response variable ($N = 53\,325$ scans), with moving as the reference category. We included the following predictor variables: (1) mean fGCM concentrations for a given individual per data collection period, (2) sex (male or female), (3) standardized rank and (4) data collection period (set as dummy variables for cold-wet, hot-wet, hot-dry and cold-dry periods). We included individual ID nested within group as a random effect. In order to understand whether fGCM concentrations played a role in the trade-off between behaviours, we compared the full model to a reduced model that excluded fGCM concentrations. In doing so, we were able to assess the influence of fGCM concentrations on the probability of a behaviour being observed, which could potentially be masked by similar variation within the data collection period. We then applied the WAIC to determine which of our models gave the best fit to our data. This model was implemented in R using 'RStan' and the 'rethinking' package [45].

## 2.10. Model 3: Influence of environmental conditions on survival

We ran an additional general linear mixed model to investigate if female/juvenile mortality was predicted by environmental factors. We fitted the data with a Poisson error structure using the 'brm' function, as with model 1, with the number of female/juvenile mortalities in a given month as our response variable ($N = 30$ months). We included food availability (monthly mean NDVI), monthly mean daily temperature and days without water (number of days without water in the previous 30 days) as predictors. We included year as a random effect. We first examined the model that included interactions between our predictor variables, but none showed any meaningful influence on the response variable and were removed to improve fit. We also found VIFs > 4, with multicollinearity between the days without water and food availability variables. In order to determine which of the two variables provided a better fit to our data we ran two reduced models, which included either food availability and temperature ($model3_{food+temp}$), or days without water and temperature ($model3_{water+temp}$). The first produced a slightly better fit (WAIC of 94.00 (±10.5 s.e.) compared to 100.2 (±10.4 s.e.)), which we suspect is because the food measure also folds in the availability of water. We present the results of both models for completeness.

## 2.11. Model 4: Survival analysis of the influence of stress and environmental variables on mortality

We ran a Cox regression model with time-varying covariates [52] to examine the influence of food, water and fGCM concentrations on mortality. Due to the repeated measures of the same individual, we used the 'coxme' function, which allows the inclusion of random effects within the model to account for pseudo-replication [52], as well as allowing us to test a number of time-varying and binary covariates. This function is currently unavailable within the Bayesian framework and so was run as a frequentist survival model. We included the date of each fGCM sample as a time point for each individual's status as alive or dead (binary: 0/1): this gave 12 known death events, and 346 events in total. Our predictor variables were: (1) fGCM concentrations, (2) standardized dominance rank, (3) food availability, (4) days without water (number of days without water in the previous 30 days), and (5) mean daily average temperature of the previous 2 days. We incorporated right-censored data to account for animals that were alive at the end of the study period. We included individual ID nested within group as a random effect.

**Table 2.** Coefficient estimates for model1$_{food+water}$ examining the influence of social and environmental factors on fGCM concentrations ($N = 346$). Shown are the estimate of the posterior means, standard error of the estimate of the posterior means and the 95% credible intervals (CI). Given are estimates on the main effects and the residual variation between individuals. Italics indicate that the estimate is greater than $\pm2$x the standard error and the majority of the 95% CI is either positive or negative.

| factor | estimate of posterior mean | estimate error | lower 95% CI | upper 95% CI |
|---|---|---|---|---|
| fixed effects: | | | | |
| *intercept* | *4.02* | *0.10* | *3.81* | *4.22* |
| *δ-intercept* | *0.55* | *0.03* | *0.49* | *0.61* |
| standardized rank | −0.03 | 0.11 | −0.24 | 0.19 |
| group: PT versus RST | −0.06 | 0.09 | −0.24 | 0.12 |
| group: PT versus RBM | −0.01 | 0.11 | −0.21 | 0.21 |
| group: RST versus RBM | 0.05 | 0.10 | −0.14 | 0.26 |
| sex (ref: male) | 0.10 | 0.11 | −0.11 | 0.30 |
| days without water | 0.09 | 0.13 | −0.17 | 0.35 |
| *food availability* | *−0.12* | *0.10* | *−0.33* | *0.08* |
| *interaction of days without water and food availability* | *0.50* | *0.23* | *0.05* | *0.96* |
| individual residual variance: | | | | |
| *intercept* | *0.21* | *0.08* | *0.06* | *0.36* |
| *standardized rank* | *0.34* | *0.16* | *0.05* | *0.66* |
| days without water | 0.11 | 0.08 | 0.01 | 0.31 |
| food availability | 0.11 | 0.08 | 0.00 | 0.30 |
| interaction of days without water and food availability | 0.38 | 0.26 | 0.02 | 0.98 |

In order to check the validity of our cox model we examined the Cox–Snell residuals, scaled Schoenfeld residuals using the function 'cox.zph' in the R package 'survival' [53] and, for influential cases, used df beta to look at score residuals. One influential case was discovered showing a large influence on the water availability variable and removed from the analysis [52]. We examined VIFs derived from a binomial model containing all predictor variables to assess collinearity and found VIFs were less than 4 indicating no collinearity. $R^2$ values for the full models were calculated using the function 'r.squaredLR' [47].

# 3. Results

## 3.1. Influence of environmental conditions on fGCM concentrations

In total, we analysed 346 faecal samples for fGCM concentrations across the time periods (table 1). This gave an average of $5.86 \pm 1.98$ samples (mean ± s.d., range 2–8) per individual ($N = 59$ individuals, not all individuals were included in all time periods due to migrations, deaths and less than two samples being collected for that individual in the time period).

In line with our prediction concerning reactive homeostasis, we found a meaningful positive interaction between food availability and days without water (table 2, figure 2). To depict the interaction graphically, we split the days without water variable into three categories: none (no water available), some days (the mid-point in our data: water available for 24 days only) and all days (water fully available; figure 2). When water was readily available in the home range and food availability was high, fGCM concentrations were lower than when food availability was high but there was a complete lack of standing water in the home range. When food availability was low, fGCM concentrations did not appear to be influenced any further by water availability. We also found Model1$_{food+water}$ revealed an overall meaningful negative effect of food availability on fGCM

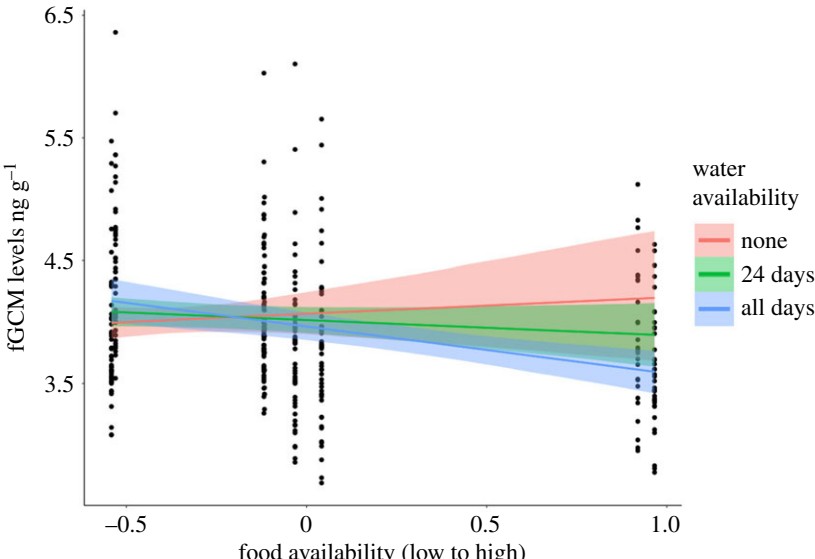

**Figure 2.** Interaction of number of days without water and variation in food availability on fGCMs (Model1$_{food+water}$; $N = 346$). Water availability is split into (1) none (no water available in the previous 30 days; red line), (2) some days (mean value: water available for 24 of the previous 30 days; green line, this represents the mean score for this variable) and all days (water available on all of the previous 30 days; blue line). Food availability is measured as the NDVI score of the previous 14 days. Food availability is z-transformed. Shown are the marginal effects of the interaction of food and water availability on log-transformed fGCM concentrations in nanograms per gram (y-axis). These categories were used only for illustrative purposes; water availability was entered as a continuous variable in all models.

concentrations: as food availability decreased, fGCM concentrations increased. As this variable was included in the interaction term, however, this result should be interpreted with caution.

The full model explained 21.7% of the variance, with the main effects explaining 7.7% of the variance. Examination of the marginal $R^2$ without the random slopes explained 16.1% of the variance, which indicates that the residual variance within the random slopes for individual identity explained 5.6% of the variance (table 2). The random slopes suggest some variation in the way individual fGCM concentrations respond to changes in standardized dominance rank but against prediction, not in response to ecological variables.

## 3.2. Trade-off of behaviours under different environmental conditions

In order to investigate any differences in the behaviours expressed by individuals under different environmental conditions, we ran a multilevel multinomial behaviour model (model2$_{behaviour}$). Firstly, we ran a model containing mean fGCM concentrations per month/individual to examine if fGCM concentrations predicted the probability of a behaviour being expressed. Comparison of this model to a reduced model without the fGCM variable showed similar WAICs and a lack of meaningful differences (WAIC$_{FULLMODEL}$ = 133995.9 (±202.52 s.e.), WAIC$_{REDUCEDMODEL}$ = 133991.3 (±202.64 s.e.)). Thus, differences in fGCM concentrations did not affect the probability of any one of the four behavioural categories being expressed over any other. There was, however, a meaningful influence of time period on the probability that certain behaviours would be expressed: time spent resting was much higher in the hot-dry period. This behaviour was traded off against reduced feeding and moving for females, and with moving for males. Social behaviour also declined through time with approximately 5% less social behaviour observed in the last cold-dry period compared to the earlier cold-wet period (table 3, figure 3). We also found that females engaged in more social behaviour and rested less than males. The latter also moved more than females in general (figure 3). Standardized rank showed no meaningful influence on the probability of any behaviour being expressed (electronic supplementary material, table S2). The full model output and summary can be found in the electronic supplementary material (electronic supplementary material, table S2).

## 3.3. Influence of environmental conditions on survival

We recorded 46 deaths between January 2014 and July 2017, 52% of which occurred between October 2016 and March 2017 (table 4). This includes the hot-dry period when no water was available in the

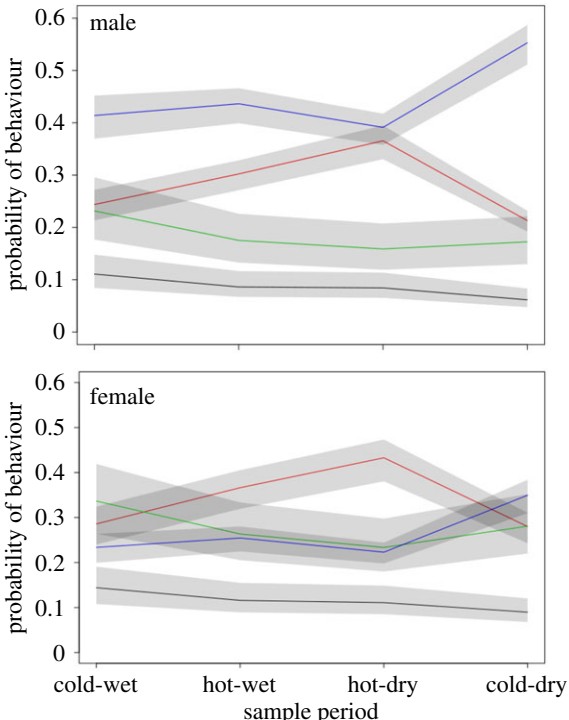

**Figure 3.** The relationship between the probability of one of four behaviours being expressed in each of the four time periods (cold-wet, hot-wet, hot-dry and cold-dry) for males and females. The four behaviours are: (1) social (black line), (2) resting (red line), (3) foraging (green line) and (4) moving (blue line). The grey bars around the lines are the 95% credibility intervals. The y-axis indicates the probability of each of the four behaviours occurring during each of the four time periods ranging between 1 (very likely to occur) and 0 (not likely to occur at all).

**Table 3.** Showing the mean probability of each of the four behavioural activities for both males and females from the multinomial behaviour model (model2$_{behaviour}$; $N = 53\,325$ scans). Given is the mean probability of each behaviour occurring in the given time period (cold-wet, hot-wet, hot-dry and cold-dry) and the 95% credible interval (CI) in parenthesis from 1000 iterations.

| sex | behaviour | mean probability of behaviour (95% CI) | | | |
|---|---|---|---|---|---|
| | | cold-wet | hot-wet | hot-dry | cold-dry |
| male | social | 0.11 (0.07–0.16) | 0.09 (0.05–0.12) | 0.08 (0.05–0.12) | 0.06 (0.04–0.09) |
| | resting | 0.24 (0.21–0.28) | 0.30 (0.26–0.34) | 0.37 (0.32–0.41) | 0.21 (0.19–0.24) |
| | foraging | 0.23 (0.15–0.32) | 0.18 (0.11–0.24) | 0.16 (0.10–0.22) | 0.17 (0.11–0.23) |
| | moving | 0.42 (0.36–0.47) | 0.44 (0.39–0.48) | 0.39 (0.35–0.43) | 0.55 (0.50–0.60) |
| female | social | 0.14 (0.08–0.20) | 0.12 (0.07–0.16) | 0.11 (0.07–0.16) | 0.09 (0.05–0.13) |
| | resting | 0.29 (0.23–0.34) | 0.37 (0.31–0.42) | 0.43 (0.37–0.50) | 0.28 (0.23–0.33) |
| | foraging | 0.34 (0.23–0.44) | 0.26 (0.18–0.36) | 0.23 (0.15–0.19) | 0.28 (0.20–0.38) |
| | moving | 0.23 (0.19–0.28) | 0.26 (0.22–0.29) | 0.22 (0.19–0.25) | 0.35 (0.30–0.40) |

animals' territories. Firstly, examining model3$_{food+temp}$, we found that food availability predicted the number of monthly mortalities in our vervet monkey groups: as monthly food availability decreased the likelihood of mortality within our population increased (table 5, figure 4). There was no meaningful influence of mean daily average temperature on mortality per month. Examination of model3$_{water+temp}$ showed that the number of days without water strongly and positively predicted the number of monthly mortalities in our vervet monkey groups. In this model, there was also a meaningful influence of mean daily average temperature on the number of monthly mortalities.

**Table 4.** Showing the number of mortalities per time period for all age classes and females only ($N = 46$ mortalities). Males are excluded from the totals as they often emigrate to unknown groups.

| time period | number of deaths of all individuals | number of deaths of adult females |
|---|---|---|
| Jan–Mar 2015 | 0 | 0 |
| Apr–Jun 2015 | 3 | 1 |
| July–Sept 2015 | 3 | 2 |
| Oct–Dec 2015 | 3 | 1 |
| Jan–Mar 2016 | 3 | 0 |
| Apr–Jun 2016 | 2 | 1 |
| July–Sept 2016 | 5 | 3 |
| Oct–Dec 2016 | 13 | 5 |
| Jan–Mar 2017 | 13 | 4 |
| Apr–Jun 2017 | 1 | 0 |

**Table 5.** Coefficient estimates for the Poisson models (Model3$_{food+temp}$ and model3$_{water+temp}$) examining the influence of environmental factors on mortality ($N = 30$ months). Shown are the estimate of the posterior means, standard error of the estimate of the posterior means and the 95% credible intervals (CI). Given are estimates on the main effects and the residual variation between individuals. Italics indicate that the estimate is greater than $\pm 2$x the standard error and the majority of the 95% CI is either positive or negative. Model3$_{food+temp}$: $R^2_{MARGINAL} = 0.430$, $R^2_{CONDITIONAL} = 0.461$. Model3$_{water+temp}$: $R^2_{MARGINAL} = 0.362$, $R^2_{CONDITIONAL} = 0.362$.

| model | factor | estimate of posterior mean | estimate error | lower 95% CI | upper 95% CI |
|---|---|---|---|---|---|
| food+temp | intercept | 0.060 | 0.480 | −0.960 | 1.180 |
| | daily temperature | 0.080 | 0.340 | −0.580 | 0.800 |
| | *food availability* | *−1.87* | *0.510* | *−2.920* | *−0.920* |
| water+temp | intercept | 0.180 | 0.180 | −0.180 | 0.520 |
| | *daily temperature* | *0.880* | *0.330* | *0.220* | *1.520* |
| | *days without water* | *1.11* | *0.280* | *0.550* | *1.640* |

## 3.4. Survival analysis of the influence of stress and environmental variables on mortality

There was some suggestion that fGCM concentrations had a negative influence on survivorship (model 4). As fGCM concentrations increased by 1 unit the likelihood of mortality was 1.9 times greater (hazard ratio = 1.88; $R^2 = 0.457$; table 6, figure 5). This finding, however, did not reach conventional significance, so caution is warranted here. There was no evidence for an influence of temperature or food availability on survival probability in this model.

## 4. Discussion

Our results suggest that drought conditions exert a meaningful influence on fGCM concentrations in our vervets. We found effects of seasonal food availability on fGCM concentrations, indicative of predictive homeostasis, but evidence for reactive homeostasis was not clear-cut. Under conditions when food availability was high, but there was no water in the home range, fGCM concentrations were increased relative to periods when water was fully available, suggesting that animals needed to work harder to sustain homeostasis under drought conditions. When food availability was low, however, there was no additional influence of water availability on fGCM concentrations, which did not fit with our predictions—we expected that a lack of water would exacerbate the lack of food, such that the animals would need to work harder to restore homeostasis.

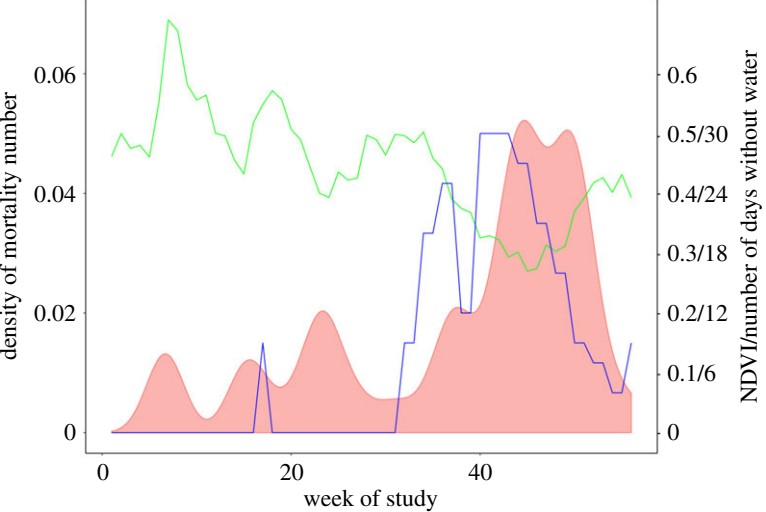

**Figure 4.** The relationship of food availability (NDVI, green line), days without water (the number of the previous 30 days water was unavailable, blue line) and the number of mortalities during the study period (model 3; density plot, red shaded area, $N = 46$ deaths). NDVI and mortality rate are calculated twice per month, which is given on the x-axis ($N = 56$ periods, $0 =$ Jan 2015, $56 =$ July 2017).

**Table 6.** Output of the Cox proportional hazards model (model 4) to investigate the influence of environmental and physiological factors on mortality ($N = 345$ time points and 12 mortalities). Whole model: log-likelihood $= -46.054$, $R^2_{adj} = 0.441$. $\beta$ is the hazard rate coefficient where a positive value indicates an increased risk of mortality.

| factor | $\beta \pm$ s.e. | hazard ratio | z value | Pr(>|z|) |
|---|---|---|---|---|
| fGCM levels | $0.631 \pm 0.344$ | 1.880 | 1.840 | 0.066 |
| daily temperature | $-1.470 \pm 1.083$ | 0.230 | $-1.360$ | 0.170 |
| food availability | $-0.071 \pm 4.355$ | 1.074 | 0.02 | 0.990 |
| standardized rank | $-0.516 \pm 0.633$ | 0.600 | $-0.820$ | 0.410 |
| water availability | $0.959 \pm 3.011$ | 2.610 | 0.320 | 0.170 |

One possible explanation for this pattern is that the animals were forced into a phase of reactive homeostasis (or possibly even homeostatic overload in some cases) throughout the drought period, and the decline seen in fGCM concentrations with increased food and water availability represents a return to the zone of predictive homeostasis. That is, our data are better characterized as capturing the animals' response to the subsequent amelioration of extremely harsh environmental conditions. This bears some resemblance to the thermal response of chacma baboons (*Papio hamadryas ursinus*) to the absence of free-standing water [54,55]. Under such conditions, it has been suggested that baboons effectively switch off evaporative sweating as a means to conserve body water, becoming increasingly hyperthermic as a result. That is, they no longer attempt to sustain homeostasis with respect to the body temperature (which could be seen as homeostatic overload in the reactive scope framework), and this is only restored once water becomes available again.

We also did not find any evidence for inter-individual differences (i.e. no influence of random effects) in reactive scope in relation to food or water availability. Possibly this is because the vervets in our study groups have all experienced similar environmental conditions across their lifespan, such that they have accumulated comparable amounts of wear-and-tear, and their reactive scopes do not differ in any meaningful fashion. This seems inherently unlikely, however, given other data from our study site. For example, we observe large differences in growth rates within and across yearly juvenile cohorts [56]. Alternatively, as suggested above, the effects of drought conditions may have pushed all individuals into the zone of reactive homostasis (or beyond) such that any inter-individual differences in fGCM concentrations were obscured by these more extreme conditions. This also seems unlikely, however, given that we observed inter-individual variation in fGCM concentrations in relation to dominance

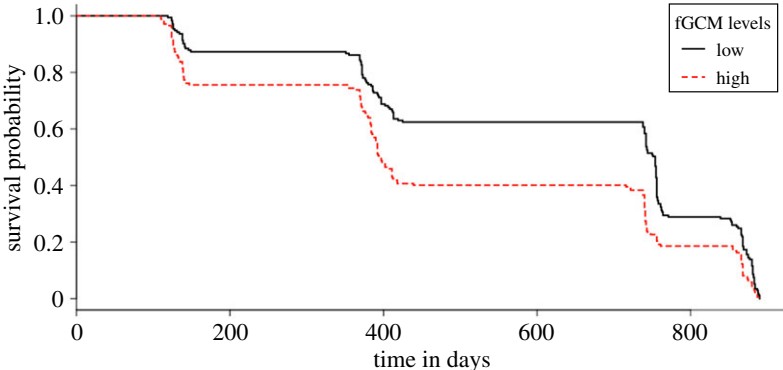

**Figure 5.** The influence of high or low fGCM concentrations on mortality (model 4). The *y*-axis indicates the probability of survival and indicated on the *x*-axis is time in days since the start of data collection. For illustrative purposes the data are split into two groups via the mean: low (lower than the mean; solid black line) and high fGCM concentrations (higher than the mean; dashed red line).

rank. The most reasonable interpretation is that our fGCM data collection was not fine-grained enough to capture inter-individual differences in response to ecological variables.

Our analyses also showed that variation in fGCM concentrations did not predict whether certain behaviours were more likely to be expressed over any others. As expected, however, we did find that animals were more likely to rest during the hot-dry period of our sample, with females trading this off against both time spent foraging and moving, whereas males traded off against moving time alone. This sex difference may reflect the fact that males spend absolutely more time moving than females, and so come under greater pressure to cut back on this when times are tough. We also observed effects on social time, which took the form of an overall decline across the study period.

Finally, our investigation of the link between extreme environmental conditions and mortality showed that a reduction in food availability predicted mortality in our population, as did a lack of water availability and high average temperatures. The best fitting model was the one that included food availability, rather than water availability. We suggest this occurred because the former model also captures the effect of water availability (i.e. lack of water is a clear cause of low food availability) whereas the model containing water availability alone does not capture the availability of food resources. It is clear that the highest level of mortality occurred when both food resources and water availability were at their lowest (figure 5), and we need to consider both our statistical models to make sense of our findings here. Lower food quality would also mean the vervets acquired less water from the consumption of food items, which would further exacerbate the water deficit.

In addition, the majority of adult female deaths occurred between October and March, which coincides with the birth and lactation period. The majority of females gave birth shortly before their deaths and it may be that the additional energetic costs of lactation played a major role in the mortality patterns we observed. As vervets are seasonal breeders, any reproduction-related effects would coincide with environmental effects in our study and are thus hard to tease apart. However, future studies should consider such life-history-related effects.

We also found some (albeit weak) evidence to suggest that fGCM concentrations predicted survivorship: animals with higher fGCM concentrations were less likely to survive the drought. Although there was some indication that lower temperatures may have ameliorated mortality risk, we did not find any clear influence of our climatic variables on survival. This is most likely due to small sample size (i.e. only 12 deaths out of 59 individuals). A lack of any direct climatic effects may also reflect the fact that fGCM concentrations already fold in the effect of the animals' response to environmental conditions. Thus, there is no additional direct influence of each climatic variable on survival, because the response to climate is already reflected in each individual's fGCM output.

Although previous studies have shown elevated physiological stress in relation to harsh environmental conditions [5,11,30,57,58], studies providing evidence of a link between fGCM concentrations and survival are rare, especially in long-lived species, such as non-human primates [57]. Only one other non-human primate study has found that elevated fGCM concentrations (weakly) predicted mortality [59]. In another social mammal, the yellow-bellied marmot (*Marmota flaviventris*), elevated baseline fGCM concentrations predicted mortality in the following year [60]. However, Bonier *et al.* [61] did not find any conclusive evidence that elevated baseline glucocorticoid levels led

to negative fitness outcomes collating studies on mammals, fish, reptiles and birds. Here, we add to this body of literature showing, on the one hand, that extreme environmental stressors lead to higher fGCM concentrations and, on the other, presenting some suggestive evidence that higher fGCM concentrations may increase the likelihood of mortality in a wild vervet monkey population.

Overall, our results show that harsh environmental conditions are associated with higher fGCM concentrations in ways that are consistent with the reactive scope model, and that mortality is increased under the most extreme drought conditions. During the period of lowest food and water availability, it seems that some animals were pushed into homeostatic overload, and became physiologically unable to sustain homeostasis. Once conditions improved, fGCM concentrations dropped, but our suggestion is that those individuals that survived nevertheless sustained wear-and-tear (i.e. suffered an increased allostatic load), which may increase their vulnerability to any future droughts. It is also apparent that inter-individual variation in fGCM concentrations, and hence differences in reactive scope, were associated with dominance rank, apparently independently of any environmental effects; something deserving of further detailed investigation.

Ethics. All protocols were non-invasive and adhered to the laws and guidelines of South Africa and Canada. Procedures were approved by the University of Lethbridge Animal Welfare Committee (Protocols 0702 and 1505). This study also adheres to the Association for the Study of Animal Behaviour/Animal Behavior Society Guidelines for the Use of Animals in Research. It is important to note that this study was carried out on wild habituated vervet monkeys under natural conditions and non-invasively. As a result no animals were harmed during the research and any mortality reported occurred naturally. The Tompkins family, owners of the Samara Private Game Reserve, granted permission to conduct the study on their property in South Africa and also approved the study.

Data accessibility. Data and R code for analysis of models 1–4 are available within the Dryad Digital Repository: https://dx.doi.org/10.5061/dryad.8jm8496 [62].

Authors' contributions. C.Y., L.R.B., S.K., R.M., S.P.H. and L.B. conceived the ideas and designed methodology; C.Y. and M.J.D. collected the data; C.Y., T.R.B., M.J.D. and A.G. analysed the data; C.Y., R.M., S.P.H. and L.B. led the writing of the manuscript. All authors contributed critically to the drafts, gave final approval for publication and agree to be held accountable for the work performed therein.

Competing interests. We declare we have no competing interests.

Funding. This research was funded by NRF (South Africa) awards to S.P.H., NSERC (Canada) Discovery Grant awards to S.P.H. and L.B., the Canada Research Chair program to L.B., and C.Y. was supported by a Senior Post-doctoral Fellowship at the University of Pretoria.

Acknowledgements. We are grateful to the Tompkins family for permission to work at Samara Private Game Reserve. We are indebted to all students and research assistants that assisted with data collection and to Kitty and Richard Viljoen for logistical support. We are grateful to the Endocrine Research Laboratory for their help with the laboratory work at UP. Finally, we would like to thank Tilana and Rudi Backeberg of Themela Farm Stall, Jansenville—their *Die Beste* Kudu pies are a staple of our research project. We thank the Associate Editor, Dr Alecia Carter, as well as Dr Julie Duboscq and one anonymous reviewer for insightful comments, which improved our manuscript.

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
