## [Reviewer comments · Royal Society Open Science]

Review History

RSOS-191078.R0 (Original submission)

Review form: Reviewer 1 (Patrick J. Tkaczynski)

Is the manuscript scientifically sound in its present form?

Yes

Are the interpretations and conclusions justified by the results?

Yes

Is the language acceptable?

Yes

Do you have any ethical concerns with this paper?

No

Have you any concerns about statistical analyses in this paper?

No

Recommendation?

Major revision is needed (please make suggestions in comments)

Comments to the Author(s)

Comments to the Author(s)

This paper examines the link between an extreme climatic event, hormonal stress (faecal glucocorticoid metabolites), mortality and survival in a population of wild vervet monkeys. The study uses the reactive scope model to explain any patterns between periods of drought, elevated stress and survival outcomes in these monkeys. Under such a framework, sustained elevated stress levels caused by either elevated temperature, lack of food or water (or these factors in combination) are expected to lead to deleterious effects on the health with ultimate implications for survival of individuals during this period of extreme weather.

Given the relative paucity of studies directly exploring the link between climate change, health and survival outcomes in wild animals, this is a timely and important study of relevance to a number of fields within biology and thus Open Science's readership. Overall, the paper is well written, much of the analysis clear and the results in line with certain predictions. However, I have some concerns about the focus on the reactive scope model given the dataset available and some concerns regarding the different datasets/timeframes used in different models, which should just require clarification. I outline these concerns and other minor corrections below.

Line 25: Italicise scientific name?

Line 42: Remove capitalisation of "Change"

Line 48: Delete "and" in "including heat waves, and cold snaps,"

Line 70: Not sure the Pride (2005) reference here is the one intended – there were two papers that year from Pride, one showing effects of competition on stress in lemurs, the other showing effect of stress on survival in the population.

Line 74-124: A good deal of the introduction is given over the reactive scope model, which conceives the stress response as involving multiple components (predictive homeostasis, reactive homeostasis, homeostatic overload etc.). My concern with devoting so much emphasis to this framework is the difficulty or perhaps impossibility, given the described dataset, of establishing these different components in your study population. For example, Romero et al (2012) used the peak stress response to a weak stressor to determine when homeostatic overload was occurring in their study population – how can such an interpretation be made in your study population?

While your results clearly show that the drought affected stress levels and survival, it is not clear that this rate of mortality was extreme nor that physiological stress levels in the vervets reached homeostatic overload levels or indeed pathological levels (they could be a plastic adjustment to drought events that may be more frequent than the timeframe of the study suggests).

I think the reactive scope model is useful in the interpretation of your results, but it seems a little misleading to claim that the data collection or analysis were conducted to explicitly test the model here. Alternatively, to help the reader, if you believe the different models you present represent different components of the reactive scope model, they should be presented as so.

Lines 98-124: Some references where the reactive scope model has been empirically tested would be useful here, e.g. Durant et al (2016).

Line 166: Number of adults needed here as these are the subjects in most of the models.

Line 173: Standardised how?

Lines 174-176 and throughout: The different time periods of the different variables and models is a little confusing and could do with more clarity. Here, if you need 3 months to create a reliable winner/loser matrix, how do you assign rank for every 6 week period? Or do you mean you need 3 months burn-in period before you can start assigning rank every 6 weeks?

Lines 192-199: Again the time periods and sampling regime are difficult to follow. Were the behavioural data collected continuously outside of the faecal sampling periods? Were faecal samples collected in between the four six-week periods? Were these periods chosen to minimise the cost of hormone analysis and target periods of interest? If so, some justification of these time

periods would be useful, i.e. do they represent the extremes of these climate types (cold-wet, hot-wet etc.)?

Line 198: Bi-monthly is used a few times throughout the paper – as this can mean either twice a month or every two months, different terminology would be clearer. I assume it means twice a month, otherwise within a 6-week period you could only include one sample per individual on average.

Line 217: Might be worth defining “globe temperature” – I had to look it up.

Line 218: The time lag for steroid excretion into faeces in these monkeys (and I think this population) is 24-36 hours – why would 10 days prior be relevant? Is 5 days prior likely reflected in the faecal samples? Why not use the climate data you have within the established time lag for the species?

Line 273-278: Why is group ID not included in this model? Presumably the effect of climate is generally going to be related to competition, so larger groups might be worse off (or maybe smaller, less competitive groups). Also, here, given the different physiological demands of the sexes (highlighted in the discussion), would interactions between sex and the climate variables be informative? Finally, would time period not be an important variable here? Determining differences and slopes between different time periods might be informative for framing results in the reactive scope model.

Line 347: With only three levels for the variable of group identity, it would be better to include this as a control variable in this (and indeed all models).

Line 360: What is meant by “whole models”?

Line 377: Sampling rates per time period per individual would be informative.

Table 1/line 711: This table would be better in time order to illustrate the gradual increase in GCs over time, peaking in hot-dry before a slight decline. Table would then reflect the figures better.

Line 379/447-448: Can you interpret single effects when the interaction is meaningful?

Line 393: Typo for “concentrations”

Line 407: Meaningful influence of time period?

Line 420: “hot summer period” – better to be consistent in terminology for the different time periods if this is what you are referring to here.

Line 421-429: How do you interpret temp being meaningful in one of these models and not the other?

Line 432: Phrasing is confusing, should it not be “negative influence on survivorship” or “positive influence on probability to die”?

Line 475: Jarret et al not in reference list

Line 500: “lactation played a major role in the mortality patterns”.

Line 531: “Here, we add...”

Line 534: “... populations.”

Line 545: I wonder if more exploration of this result is merited. It is interesting that there is inter-individual variation in plasticity to social stressors but not climatic stressors, especially as the social time budget remains fairly constant throughout the study in spite of challenging climate. What does this say about the significance of social vs climatic environment to physiology in this species?

Line 583: Reference lacks volume or page numbers.

Line 692: This Young et al appears after the other in main body text, so needs to be Young et al 2017b rather than 2017a.

Line 737: No bold highlighting apparent in the table

Figure 2: X-axis appears to be standardised by this is not stated in the figure or its legend.

Review form: Reviewer 2 (Julie Duboscq)

Is the manuscript scientifically sound in its present form?

Yes

Are the interpretations and conclusions justified by the results?

Yes

Is the language acceptable?

Yes

Do you have any ethical concerns with this paper?

No

Have you any concerns about statistical analyses in this paper?

No

Recommendation?

Accept with minor revision (please list in comments)

Comments to the Author(s)

Review ms RS0S-191078 Climate induced stress and mortality in vervet monkeys
by Julie Duboscq julie.duboscq@mnhn.fr

This study investigates the link between physiological and behavioural patterns of vervet monkeys and extreme environmental conditions such as drought, low food availability and high temperatures in the framework of the reactive scope model. The study is conducted on 3 different groups of monkeys living in a game reserve in South Africa. Results show that monkeys have more elevated fGCM during harsh periods of low food and water availability, that they change their behaviour accordingly, that there is a positive relationship between harsh conditions and high mortality in females and juveniles, and that higher fGCMs were associated with an increased probability of mortality. This is an important study to assess how animals deal with extreme environmental conditions which will become more frequent in the future due to climate change. Overall, this is a well-designed study with good foundations, clear objectives and masterful analyses. The only serious issue might concern the way food availability has been assessed and considered which the authors should address clearly. More details on how things were done are also needed here and there.

I know it is by no means the authors' fault and this is probably more of a message to the RS journals, I just want to remark that having the figures and tables + their legends not imbedded within the text makes for a difficult read hence assessment of the paper... isn't there an option to send stuff all tied together?

Kudos to the authors for sharing their dataset and R codes! What would be even more awesome would be 1/ (more detailed) annotated scripts and 2/ an explanatory note for each table/sheet of the dataset. For instance, in the model 2 sheet, what are Month1, Month2, Month3 referring too? I'm guessing it is the dummy coding of the period variable? Otherwise, it is not enough to fully replicate the analyses.

Line 72: How about being a bit more specific in this example by explicitly linking drought to starvation to death?

Lines 74-124: great explanations but not very concise. It might be a bit unusual in the introduction but how about adding a schematic/graph here to make those paragraphs a bit shorter and perhaps clearer?

Line 126: how do we know already that the drought period had an impact on vervets' "physiological stress levels, activity and survival probabilities"? is there a verb missing here maybe to state that "we take advantage of ... to investigate/study/test ..."?

Line 133: what is "standing water"?

Line 134: noticed here but may have to be defined earlier: what's a drought?

Lines 172 and § fwd: the readers of RSOS might not be familiar with the assessment of dominance rank in animals so it needs to be specified which behaviours are used, how data are collected, processed and arranged to be able to calculate DS... what are DSs? What's a winner/loser matrix? How to build a hierarchy from dominance ranks? "...standardized David's Scores" why and how to do that?

Line 197: this makes 332 samples but there seems to be 346 fGCM values. Why the discrepancy?

Lines 198 and § fwd: it might be useful to specify that fecal samples are individually collected, i.e. from identified individuals observed pooping. Lines 201-202: directly where the feces were deposited? On a leaf? Ground? Piece of plastic?... line 203: "during" instead of "until"? Lines 205-212: how about fleshing out a bit more how those analyses were conducted? A short summary would be nice.

Lines 227-228: it is not clear how this assessment is made, e.g., if water availability is scored as "absent" does it mean the river is gone too? Is there any data on water consumption by the monkeys? Of course, if water is totally absent of their home range, they literally cannot drink but there must be a gradient of water availability influencing their drinking behavior?

Line 229: why thirty days here when temperature is tested at a 5- and 10-days range?

Line 232 and § fwd: I get that this is all research-in-progress and that a single project results in many publications with more or less related or similar (or exact same) methodologies but for a 1st time reader, it is hard to read that x,y,z methods was used, read papers x,y,z to understand how it is done. It is ok to do so if a summary of the method is given then other publications are referenced for MORE details (emphasis on more, some details are necessary first). As for endocrinological analyses lines 205-212, it would be helpful to have some details here too for instance how is the amount of biomass measured (colour gradient? Luminosity? Light reflection?...)? what does "more abundant in biomass" mean in terms of food availability and how and why is this really a good proxy? The vervets are not folivorous are they? (How) Does foliage correlate with fruiting? Was there any test of this relationship? Maybe this all seems fairly obvious but it would be better to spit it out.

Lines 259-261: what was the rational behind those numbers, apart from it is used/proposed as settings in the papers cited, in short?

Line 269: individual fGCM concentration per fecal sample? Per 6-weeks periods? In the xls file, there's between 2 and 8 values per individual monkey, what do they represent? Wouldn't it be important to control for either number of samples per individuals or date of collection or is that implicit in controlling for ID and testing for environmental variables?

Line 270 2): the NDVI index then?

Lines 273-277: what was the rationale behind the choice of interactions? Why is dominance rank used as a control factor, to control what exactly? And what are random slopes and intercepts for?

Line 289: how about a short explanation of this criterion? E.g. (how) Is it related to the other(s) AIC? Why this one?

Lines 298 and § fwd: isn't it necessary to imbed behaviours into their scans? I.e. individual A moved in scan 1, rested in scan 2, moved again in scan 3 ect ect. Because there is a full list of behaviours for all individuals in each scan, right? Line 305: why was moving considered the reference category, and not e.g. the most frequent activity? If I understood correctly how the data are organized here, there are, let's say, 100 scans for individual A in period 1 and 115 in period 2, so there are 100 and 115 behaviour values respectively but 100 and 115 times the same fGCM value, 100 and 115 times the same standardized rank, ect. How does the model deal with this and is it an issue?

Lines 340 and § fwd: reading the text and looking at the Model4 datasheet, I don't manage to make sense of how this works. What are T2 and T1 in the datasheet?

Line 352: what was the issue with this influential case?

Lines 379, 380, ect: "meaningful" effect or interaction is another way of saying significant without saying significant? It can be meaningful without being significant and the other way around so use another term, or no qualifier at all, or stick with significant.

Lines 380-381: it would be better to describe the effect of the interaction first then of the single variable.

Lines 407 and § fwd: I don't understand the influence of time here? How does the model tell that variable in period x is higher/lower than variable in period y? Wouldn't this all be within normal variation?

Lines 424-425 and 428-429: How can this be? Maybe a sentence or 2 of interpretation would already make sense here?

Line 457: the or these animals instead of our animals

Line 478 and § above: or the timescale of analysis is not fine enough to capture individual differences?

Line 503 and § above: so, maybe it would have been judicious to include other individual factors, such as "reproductively active" i.e. engaged in sexual activities vs "non-active", "pregnant" vs "nonpregnant", "lactating" vs "nonlactating"?

Line 544: it is weird to mention this just now and here and not discuss it anywhere else, like § starting line 468 maybe or line 504.

Table 1: I would also add the dates of those periods in the first column, just as a reference.

Table 2 (and elsewhere): "certainty" intervals? The text says "credible" intervals line 255. Are they one and the same? Then consistency in term use is warranted I guess.

Table 3: can the periods be labeled consistently here too? either with a date or with a qualifier like in table 1 or both but not one or the other.

Figure 1: it would also be helpful to highlight in this graph that the purple fecal sample collection periods are also the analysed periods, i.e. "cold-wet" (April-June 2015), "hot-wet" (December 2015-February 2016), "hot-dry" (December 2016-February 2017) and "cold-dry" (April-June 2017). Furthermore what NVDI values are used here? A monthly average from the bi-monthly downloaded images? The value "relating to the closest image prior to fecal sample collection" (line 242)? Indicate again sample size and which model it does refer to.

Figure 2: why 24 out of 30? There's a typo in food availability. Indicate again sample size and which model it does refer to.

Figure 3: it would be helpful to specify in the legend what a probability of 01 or 06 means (y-axis) and what the grey area means too. Indicate again sample size and which model it does refer to.

Figure 4: x-axis title says "week of study" but legend says "periods" which if I understood correctly last 2 weeks bc bi-monthly frequency, right? So it's a bit incorrect. Indicate again sample size and which model it does refer to.

Figure 5: what's the threshold for the split? Indicate again which individual data make that plot, i.e. females and juveniles, and which model it does refer to.

Decision letter (RSOS-191078.R0)

31-Jul-2019

Dear Dr Young,

The editors assigned to your paper ("Climate induced stress and mortality in vervet monkeys") have now received comments from reviewers. We would like you to revise your paper in accordance with the referee and Associate Editor suggestions which can be found below (not including confidential reports to the Editor). Please note this decision does not guarantee eventual acceptance.

Please submit a copy of your revised paper before 23-Aug-2019. Please note that the revision deadline will expire at 00.00am on this date. If we do not hear from you within this time then it will be assumed that the paper has been withdrawn. In exceptional circumstances, extensions may be possible if agreed with the Editorial Office in advance. We do not allow multiple rounds of revision so we urge you to make every effort to fully address all of the comments at this stage. If deemed necessary by the Editors, your manuscript will be sent back to one or more of the original reviewers for assessment. If the original reviewers are not available, we may invite new reviewers.

- Data accessibility

If you wish to submit your supporting data or code to Dryad (<http://datadryad.org/>), or modify your current submission to dryad, please use the following link:
<http://datadryad.org/submit?journalID=RSOS&manu=RSOS-191078>

- Competing interests

- Authors' contributions

- Acknowledgements

- Funding statement

on behalf of Dr Alecia Carter (Associate Editor) and Kevin Padian (Subject Editor)
 openscience@royalsociety.org

Associate Editor's comments (Dr Alecia Carter):

I have now received two reviews and read your manuscript myself. The reviewers and myself agree that this study is timely, well executed and interesting. The manuscript is clear and well-written. The reviewers have provided a set of constructive and helpful comments that I hope will improve the manuscript when addressed.

Reviewers' Comments to Author:

Reviewer: 1

This paper examines the link between an extreme climatic event, hormonal stress (faecal glucocorticoid metabolites), mortality and survival in a population of wild vervet monkeys. The study uses the reactive scope model to explain any patterns between periods of drought, elevated stress and survival outcomes in these monkeys. Under such a framework, sustained elevated stress levels caused by either elevated temperature, lack of food or water (or these factors in combination) are expected to lead to deleterious effects on the health with ultimate implications for survival of individuals during this period of extreme weather.

Given the relative paucity of studies directly exploring the link between climate change, health and survival outcomes in wild animals, this is a timely and important study of relevance to a number of fields within biology and thus Open Science's readership. Overall, the paper is well written, much of the analysis clear and the results in line with certain predictions. However, I have some concerns about the focus on the reactive scope model given the dataset available and some concerns regarding the different datasets/timeframes used in different models, which should just require clarification. I outline these concerns and other minor corrections below.

Line 25: Italicise scientific name?

Line 42: Remove capitalisation of "Change"

Line 48: Delete "and" in "including heat waves, and cold snaps,"

Line 70: Not sure the Pride (2005) reference here is the one intended – there were two papers that year from Pride, one showing effects of competition on stress in lemurs, the other showing effect of stress on survival in the population.

Line 74-124: A good deal of the introduction is given over the reactive scope model, which conceives the stress response as involving multiple components (predictive homeostasis, reactive homeostasis, homeostatic overload etc.). My concern with devoting so much emphasis to this framework is the difficulty or perhaps impossibility, given the described dataset, of establishing these different components in your study population. For example, Romero et al (2012) used the peak stress response to a weak stressor to determine when homeostatic overload was occurring in their study population – how can such an interpretation be made in your study population?

While your results clearly show that the drought affected stress levels and survival, it is not clear that this rate of mortality was extreme nor that physiological stress levels in the vervets reached homeostatic overload levels or indeed pathological levels (they could be a plastic adjustment to drought events that may be more frequent than the timeframe of the study suggests).

I think the reactive scope model is useful in the interpretation of your results, but it seems a little misleading to claim that the data collection or analysis were conducted to explicitly test the model here. Alternatively, to help the reader, if you believe the different models you present represent different components of the reactive scope model, they should be presented as so.

Lines 98-124: Some references where the reactive scope model has been empirically tested would be useful here, e.g. Durant et al (2016).

Line 166: Number of adults needed here as these are the subjects in most of the models.

Line 173: Standardised how?

Lines 174-176 and throughout: The different time periods of the different variables and models is a little confusing and could do with more clarity. Here, if you need 3 months to create a reliable winner/loser matrix, how do you assign rank for every 6 week period? Or do you mean you need 3 months burn-in period before you can start assigning rank every 6 weeks?

Lines 192-199: Again the time periods and sampling regime are difficult to follow. Were the behavioural data collected continuously outside of the faecal sampling periods? Were faecal samples collected in between the four six-week periods? Were these periods chosen to minimise the cost of hormone analysis and target periods of interest? If so, some justification of these time periods would be useful, i.e. do they represent the extremes of these climate types (cold-wet, hot-wet etc.)?

Line 198: Bi-monthly is used a few times throughout the paper – as this can mean either twice a month or every two months, different terminology would be clearer. I assume it means twice a month, otherwise within a 6-week period you could only include one sample per individual on average.

Line 217: Might be worth defining “globe temperature” – I had to look it up.

Line 218: The time lag for steroid excretion into faeces in these monkeys (and I think this population) is 24-36 hours – why would 10 days prior be relevant? Is 5 days prior likely reflected in the faecal samples? Why not use the climate data you have within the established time lag for the species?

Line 273-278: Why is group ID not included in this model? Presumably the effect of climate is generally going to be related to competition, so larger groups might be worse off (or maybe smaller, less competitive groups). Also, here, given the different physiological demands of the sexes (highlighted in the discussion), would interactions between sex and the climate variables be informative? Finally, would time period not be an important variable here? Determining differences and slopes between different time periods might be informative for framing results in the reactive scope model.

Line 347: With only three levels for the variable of group identity, it would be better to include this as a control variable in this (and indeed all models).

Line 360: What is meant by “whole models”?

Line 377: Sampling rates per time period per individual would be informative.

Table 1/line 711: This table would be better in time order to illustrate the gradual increase in GCs over time, peaking in hot-dry before a slight decline. Table would then reflect the figures better.

Line 379/447-448: Can you interpret single effects when the interaction is meaningful?

Line 393: Typo for “concentrations”

Line 407: Meaningful influence of time period?

Line 420: “hot summer period” – better to be consistent in terminology for the different time periods if this is what you are referring to here.

Line 421-429: How do you interpret temp being meaningful in one of these models and not the other?

Line 432: Phrasing is confusing, should it not be “negative influence on survivorship” or “positive influence on probability to die”?

Line 475: Jarret et al not in reference list

Line 500: "lactation played a major role in the mortality patterns".

Line 531: "Here, we add..."

Line 534: "... populations."

Line 545: I wonder if more exploration of this result is merited. It is interesting that there is inter-individual variation in plasticity to social stressors but not climatic stressors, especially as the social time budget remains fairly constant throughout the study in spite of challenging climate. What does this say about the significance of social vs climatic environment to physiology in this species?

Line 583: Reference lacks volume or page numbers.

Line 692: This Young et al appears after the other in main body text, so needs to be Young et al 2017b rather than 2017a.

Line 737: No bold highlighting apparent in the table

Figure 2: X-axis appears to be standardised by this is not stated in the figure or its legend.

Reviewer: 2

Comments to the Author(s)

Review ms RS05-191078 Climate induced stress and mortality in vervet monkeys

by Julie Duboscq julie.duboscq@mnhn.fr

This study investigates the link between physiological and behavioural patterns of vervet monkeys and extreme environmental conditions such as drought, low food availability and high temperatures in the framework of the reactive scope model. The study is conducted on 3 different groups of monkeys living in a game reserve in South Africa. Results show that monkeys have more elevated fGCM during harsh periods of low food and water availability, that they change their behaviour accordingly, that there is a positive relationship between harsh conditions and high mortality in females and juveniles, and that higher fGCMs were associated with an increased probability of mortality. This is an important study to assess how animals deal with extreme environmental conditions which will become more frequent in the future due to climate change. Overall, this is a well-designed study with good foundations, clear objectives and masterful analyses. The only serious issue might concern the way food availability has been assessed and considered which the authors should address clearly. More details on how things were done are also needed here and there.

I know it is by no means the authors' fault and this is probably more of a message to the RS journals, I just want to remark that having the figures and tables + their legends not imbedded within the text makes for a difficult read hence assessment of the paper... isn't there an option to send stuff all tied together?

Kudos to the authors for sharing their dataset and R codes! What would be even more awesome would be 1/ (more detailed) annotated scripts and 2/ an explanatory note for each table/sheet of the dataset. For instance, in the model 2 sheet, what are Month1, Month2, Month3 referring too? I'm guessing it is the dummy coding of the period variable? Otherwise, it is not enough to fully replicate the analyses.

Line 72: How about being a bit more specific in this example by explicitly linking drought to starvation to death?

Lines 74-124: great explanations but not very concise. It might be a bit unusual in the introduction

but how about adding a schematic/graph here to make those paragraphs a bit shorter and perhaps clearer?

Line 126: how do we know already that the drought period had an impact on vervets' "physiological stress levels, activity and survival probabilities"? is there a verb missing here maybe to state that "we take advantage of ... to investigate/study/test ..."?

Line 133: what is "standing water"?

Line 134: noticed here but may have to be defined earlier: what's a drought?

Lines 172 and § fwd: the readers of RSOS might not be familiar with the assessment of dominance rank in animals so it needs to be specified which behaviours are used, how data are collected, processed and arranged to be able to calculate DS... what are DSs? What's a winner/loser matrix? How to build a hierarchy from dominance ranks? "...standardized David's Scores" why and how to do that?

Line 197: this makes 332 samples but there seems to be 346 fGCM values. Why the discrepancy?

Lines 198 and § fwd: it might be useful to specify that fecal samples are individually collected, i.e. from identified individuals observed pooping. Lines 201-202: directly where the feces were deposited? On a leaf? Ground? Piece of plastic?... line 203: "during" instead of "until"? Lines 205-212: how about fleshing out a bit more how those analyses were conducted? A short summary would be nice.

Lines 227-228: it is not clear how this assessment is made, e.g., if water availability is scored as "absent" does it mean the river is gone too? Is there any data on water consumption by the monkeys? Of course, if water is totally absent of their home range, they literally cannot drink but there must be a gradient of water availability influencing their drinking behavior?

Line 229: why thirty days here when temperature is tested at a 5- and 10-days range?

Line 232 and § fwd: I get that this is all research-in-progress and that a single project results in many publications with more or less related or similar (or exact same) methodologies but for a 1st time reader, it is hard to read that x,y,z methods was used, read papers x,y,z to understand how it is done. It is ok to do so if a summary of the method is given then other publications are referenced for MORE details (emphasis on more, some details are necessary first). As for endocrinological analyses lines 205-212, it would be helpful to have some details here too for instance how is the amount of biomass measured (colour gradient? Luminosity? Light reflection?...)? what does "more abundant in biomass" mean in terms of food availability and how and why is this really a good proxy? The vervets are not folivorous are they? (How) Does foliage correlate with fruiting? Was there any test of this relationship? Maybe this all seems fairly obvious but it would be better to spit it out.

Lines 259-261: what was the rational behind those numbers, apart from it is used/proposed as settings in the papers cited, in short?

Line 269: individual fGCM concentration per fecal sample? Per 6-weeks periods? In the xls file, there's between 2 and 8 values per individual monkey, what do they represent? Wouldn't it be important to control for either number of samples per individuals or date of collection or is that implicit in controlling for ID and testing for environmental variables?

Line 270 2): the NDVI index then?

Lines 273-277: what was the rationale behind the choice of interactions? Why is dominance rank used as a control factor, to control what exactly? And what are random slopes and intercepts for?

Line 289: how about a short explanation of this criterion? E.g. (how) Is it related to the other(s) AIC? Why this one?

Lines 298 and § fwd: isn't it necessary to imbed behaviours into their scans? I.e. individual A moved in scan 1, rested in scan 2, moved again in scan 3 ect ect. Because there is a full list of behaviours for all individuals in each scan, right? Line 305: why was moving considered the reference category, and not e.g. the most frequent activity? If I understood correctly how the data are organized here, there are, let's say, 100 scans for individual A in period 1 and 115 in period 2, so there are 100 and 115 behaviour values respectively but 100 and 115 times the same fGCM value, 100 and 115 times the same standardized rank, ect. How does the model deal with this and is it an issue?

Lines 340 and § fwd: reading the text and looking at the Model4 datasheet, I don't manage to make sense of how this works. What are T2 and T1 in the datasheet?

Line 352: what was the issue with this influential case?

Lines 379, 380, ect: "meaningful" effect or interaction is another way of saying significant without saying significant? It can be meaningful without being significant and the other way around so use another term, or no qualifier at all, or stick with significant.

Lines 380-381: it would be better to describe the effect of the interaction first then of the single variable.

Lines 407 and § fwd: I don't understand the influence of time here? How does the model tell that variable in period x is higher/lower than variable in period y? Wouldn't this all be within normal variation?

Lines 424-425 and 428-429: How can this be? Maybe a sentence or 2 of interpretation would already make sense here?

Line 457: the or these animals instead of our animals

Line 478 and § above: or the timescale of analysis is not fine enough to capture individual differences?

Line 503 and § above: so, maybe it would have been judicious to include other individual factors, such as "reproductively active" i.e. engaged in sexual activities vs "non-active", "pregnant" vs "nonpregnant", "lactating" vs "nonlactating"?

Line 544: it is weird to mention this just now and here and not discuss it anywhere else, like § starting line 468 maybe or line 504.

Table 1: I would also add the dates of those periods in the first column, just as a reference.

Table 2 (and elsewhere): "certainty" intervals? The text says "credible" intervals line 255. Are they one and the same? Then consistency in term use is warranted I guess.

Table 3: can the periods be labeled consistently here too? either with a date or with a qualifier like in table 1 or both but not one or the other.

Figure 1: it would also be helpful to highlight in this graph that the purple fecal sample collection periods are also the analysed periods, i.e. "cold-wet" (April-June 2015), "hot-wet" (December 2015-February 2016), "hot-dry" (December 2016-February 2017) and "cold-dry" (April-June 2017). Furthermore what NVDI values are used here? A monthly average from the bi-monthly downloaded images? The value "relating to the closest image prior to fecal sample collection" (line 242)? Indicate again sample size and which model it does refer to.

Figure 2: why 24 out of 30? There's a typo in food availability. Indicate again sample size and which model it does refer to.

Figure 3: it would be helpful to specify in the legend what a probability of 01 or 06 means (y-axis) and what the grey area means too. Indicate again sample size and which model it does refer to.

Figure 4: x-axis title says "week of study" but legend says "periods" which if I understood correctly last 2 weeks bc bi-monthly frequency, right? So it's a bit incorrect. Indicate again sample size and which model it does refer to.

Figure 5: what's the threshold for the split? Indicate again which individual data make that plot, i.e. females and juveniles, and which model it does refer to.

Author's Response to Decision Letter for (RSOS-191078.R0)

See Appendix A.

Decision letter (RSOS-191078.R1)

04-Oct-2019

Dear Dr Young:

On behalf of the Editors, I am pleased to inform you that your Manuscript RSOS-191078.R1 entitled "Climate induced stress and mortality in vervet monkeys" has been accepted for publication in Royal Society Open Science subject to minor revision in accordance with the referee suggestions. Please find the referees' comments at the end of this email.

The reviewers and Subject Editor have recommended publication, but also suggest some minor revisions to your manuscript. Therefore, I invite you to respond to the comments and revise your manuscript.

- Ethics statement

- Data accessibility

It is a condition of publication that all supporting data are made available either as supplementary information or preferably in a suitable permanent repository. The data accessibility section should state where the article's supporting data can be accessed. This section should also include details, where possible of where to access other relevant research materials

such as statistical tools, protocols, software etc can be accessed. If the data has been deposited in an external repository this section should list the database, accession number and link to the DOI for all data from the article that has been made publicly available. Data sets that have been deposited in an external repository and have a DOI should also be appropriately cited in the manuscript and included in the reference list.

If you wish to submit your supporting data or code to Dryad (<http://datadryad.org/>), or modify your current submission to dryad, please use the following link:
<http://datadryad.org/submit?journalID=RSOS&manu=RSOS-191078.R1>

- **Competing interests**

- **Authors' contributions**

- **Acknowledgements**

- **Funding statement**

Because the schedule for publication is very tight, it is a condition of publication that you submit the revised version of your manuscript before 13-Oct-2019. Please note that the revision deadline will expire at 00.00am on this date. If you do not think you will be able to meet this date please let me know immediately.

on behalf of Dr Alecia Carter (Associate Editor) and Kevin Padian (Subject Editor)
openscience@royalsociety.org

Associate Editor Comments to Author (Dr Alecia Carter):
Associate Editor
Comments to the Author:

Dear authors,

Thank you for taking the time to carefully address the reviewers' comments on your earlier submission. I have only a few small grammatical corrections to suggest, outlined below. Thank you for submitting this manuscript to RSOS.

L74. Please move the Latin species name to after the mention of the species.
 L120. monkeys -> monkeys'
 L211 and elsewhere: space required between number and unit
 L222: weighted -> weighed
 L253: please split this paragraph here
 L313: a semi-colon would be more appropriate here (after the reference) than a comma
 L547: more likely -> greater (as you already mention likelihood earlier in the sentence)
 L626: morality -> mortality (though I do like the implied idea of the goodness of the mothers' ultimate sacrifices)
 L636: animal's -> animals'
 L653: in wild -> in a wild

Author's Response to Decision Letter for (RSOS-191078.R1)

See Appendix B.

Decision letter (RSOS-191078.R2)

22-Oct-2019

Dear Dr Young,

I am pleased to inform you that your manuscript entitled "Climate induced stress and mortality in vervet monkeys" is now accepted for publication in Royal Society Open Science.

Kind regards,
 Lianne Parkhouse
 Editorial Coordinator

on behalf of Dr Alecia Carter (Associate Editor) and Kevin Padian (Subject Editor)
openscience@royalsociety.org

Appendix A

Christopher Young Ph.D
Mammal Research Institute,
University of Pretoria
Faculty of Natural and Agricultural Science,
Pretoria,
Republic of South Africa
Christopher.young@uleth.ca

Ref: RSOS-191078

Pretoria, 23rd September 2019

Dear Dr. Carter,

We thank you for the opportunity to resubmit a revised version of our manuscript RSOS-191078 (*Climate induced stress and mortality in vervet monkeys*). We would also like – through you – to thank both Reviewers for their insightful and constructive comments. We appreciate your comments and the comments of the Reviewers, which we feel have substantially improved the quality of the manuscript.

In regards to the sharing of data we have after careful consideration altered the data available for model 2. This model consists of a large chunk of behavioural scan data (55,000+ scans), which are currently part of our long-term research project and part of several additional manuscripts and projects. We have instead included a subset of data (approx. 250 scans) so that readers can “try out” the code from the supplementary material online and the full data set can be made available to interested parties on request. The data for models 1,3 and 4 remains in full and is fully accessible in the online format. We hope you understand our reasoning for this change to the data accessibility and would appreciate any guidance from the Editor if a different course of action should be taken.

Please find below our detailed responses to the Reviewer comments (marked in red and italics). Thank you again for your review of our manuscript.

Yours sincerely,

Dr. Christopher Young (on behalf of my co-authors)

Associate Editor's comments (Dr Alecia Carter):

I have now received two reviews and read your manuscript myself. The reviewers and myself agree that this study is timely, well executed and interesting. The manuscript is clear and well-written. The reviewers have provided a set of constructive and helpful comments that I hope will improve the manuscript when addressed.

We thank you for your positive comments. We have addressed the comments of the reviewers below. Our responses are given in red italic type.

Reviewers' Comments to Author:

Reviewer: 1

This paper examines the link between an extreme climatic event, hormonal stress (faecal glucocorticoid metabolites), mortality and survival in a population of wild vervet monkeys. The study uses the reactive scope model to explain any patterns between periods of drought, elevated stress and survival outcomes in these monkeys. Under such a framework, sustained elevated stress levels caused by either elevated temperature, lack of food or water (or these factors in combination) are expected to lead to deleterious effects on the health with ultimate implications for survival of individuals during this period of extreme weather.

Given the relative paucity of studies directly exploring the link between climate change, health and survival outcomes in wild animals, this is a timely and important study of relevance to a number of fields within biology and thus Open Science's readership. Overall, the paper is well written, much of the analysis clear and the results in line with certain predictions. However, I have some concerns about the focus on the reactive scope model given the dataset available and some concerns regarding the different datasets/timeframes used in different models, which should just require clarification. I outline these concerns and other minor corrections below.

Thank you for your positive comments on our manuscript. We address your concerns below.

Line 25: Italicise scientific name?

This change has been made.

Line 42: Remove capitalisation of "Change"

This change has been made.

Line 48: Delete "and" in "including heat waves, and cold snaps,"

This change has been made.

Line 70: Not sure the Pride (2005) reference here is the one intended – there were two papers that year from Pride, one showing effects of competition on stress in lemurs, the other showing effect of stress on survival in the population.

We thank the reviewer for noticing this typo, we have now added the correct citation.

Line 74-124: A good deal of the introduction is given over the reactive scope model, which conceives the stress response as involving multiple components (predictive homeostasis, reactive homeostasis, homeostatic overload etc.). My concern with devoting so much emphasis to this framework is the difficulty or perhaps impossibility, given the described dataset, of establishing these different components in your study population. For example, Romero et al (2012) used the peak stress response to a weak stressor to determine when homeostatic overload was occurring in their study population – how can such an interpretation be made in your study population?

We position our study in a reactive scope framework by viewing fGCM as indicators of how hard animals have to work to sustain homeostasis. Our aim here was to use the reactive scope as our theoretical framework to examine fGCM concentrations rather than the still more common view in many areas of animal behaviour and particularly primatology that high fGCM concentrations are bad and indicate high “stress”. We understand that our explanations of the reactive scope may have been a bit detailed and as suggested by the reviewer we have reduced this part of the introduction.

While your results clearly show that the drought affected stress levels and survival, it is not clear that this rate of mortality was extreme nor that physiological stress levels in the vervets reached homeostatic overload levels or indeed pathological levels (they could be a plastic adjustment to drought events that may be more frequent than the timeframe of the study suggests).

I think the reactive scope model is useful in the interpretation of your results, but it seems a little misleading to claim that the data collection or analysis were conducted to explicitly test the model here. Alternatively, to help the reader, if you believe the different models you present represent different components of the reactive scope model, they should be presented as so.

As suggested, we have altered our introduction and predictions to better describe our aims of the paper. We are not attempting to explicitly test the reactive scope model but we are using it as a theoretical framework in which to base the explanation of our results.

Lines 98-124: Some references where the reactive scope model has been empirically tested would be useful here, e.g. Durant et al (2016).

We have now included this and additional references to the introduction.

Line 166: Number of adults needed here as these are the subjects in most of the models.

This information has been added.

Line 173: Standardised how?

For each three-month time period and each group separately the score for each individual was divided by the highest score for that period, giving the highest ranking individual a score of 1. This has been added to the text.

Lines 174-176 and throughout: The different time periods of the different variables and models is a little confusing and could do with more clarity. Here, if you need 3 months to create a reliable winner/loser matrix, how do you assign rank for every 6 week period? Or do you mean you need 3 months burn-in period before you can start assigning rank every 6 weeks?

We use three-month periods to build the winner-loser matrix as this time frame provided us with enough data points in order to construct a dominance hierarchy with few unknown relationships. Therefore, each matrix covers a three-month period e.g. Jan to Mar, Apr to Jun etc. Depending on when the faecal sample was collected, an individual could have different dominance scores e.g., if they had one sample collected on Dec 28th and one collected on Feb 2nd, then these two samples would fall within the six-week collection period but would also fall across two different three-month periods. We have revised the text to make this approach clearer.

Lines 192-199: Again the time periods and sampling regime are difficult to follow. Were the behavioural data collected continuously outside of the faecal sampling periods? Were faecal samples collected in between the four six-week periods? Were these periods chosen to minimise the cost of hormone analysis and target periods of interest? If so, some justification of these time periods would be useful, i.e. do they represent the extremes of these climate types (cold-wet, hot-wet etc.)?

Behavioural data were indeed collected continuously outside the faecal sampling periods. We collect data 5 days per week, 10 hours per day, as a standard protocol. The four windows of six-weeks were selected as a means to examine periods with distinctly different ecological conditions. These periods represent four periods of extremes of climatic conditions 1) cold-wet: winter with low temperatures but flowing river and higher NDVI; 2) hot-wet: high temperatures but flowing river and mid-high NDVI; 3) cold-dry: low temperatures, low NDVI and lack of water in the home-range; 4) hot-dry: high temperatures, low NDVI and lack of water in the home-range.

Line 198: Bi-monthly is used a few times throughout the paper – as this can mean either twice a month or every two months, different terminology would be clearer. I assume it means twice a month, otherwise within a 6-week period you could only include one sample per individual on average.

Bi-monthly refers to twice per month. We have changed the text to read “twice per month” in order to make this clearer to the reader.

Line 217: Might be worth defining “globe temperature” – I had to look it up.

This has been added to the text.

Line 218: The time lag for steroid excretion into faeces in these monkeys (and I think this population) is 24-36 hours – why would 10 days prior be relevant? Is 5 days prior likely

reflected in the faecal samples? Why not use the climate data you have within the established time lag for the species?

In general there is no consensus in the literature for which time period is relevant in terms of how temperatures can affect physiology of individuals. Therefore we selected two time periods we felt would represent short term (5 days) and long term (10 days) impacts of the monkey's physiology. And then compare the two periods statistically (Using WAIC and model comparison) to determine which gave a better fit to our data. However, as the Reviewer states we do have prior knowledge that the time lag of steroid excretion in our population is approx. 2 days and so as a result we have re-run the analysis using a two day window of average daily temperature. As we agree with the reviewer that this will more likely represent the impact of temperature on the individual.

Line 273-278: Why is group ID not included in this model? Presumably the effect of climate is generally going to be related to competition, so larger groups might be worse off (or maybe smaller, less competitive groups). Also, here, given the different physiological demands of the sexes (highlighted in the discussion), would interactions between sex and the climate variables be informative? Finally, would time period not be an important variable here? Determining differences and slopes between different time periods might be informative for framing results in the reactive scope model.

We have now included group ID as a main effect and control factor in our models. We do not look at interactions between sex and climatic variables as we include sex as a control factor in the models. We do not include time periods as a variable as due to the distinct differences in climatic variables of these four time periods. The model is designed to test the differences of climatic variables between these four time periods but on each climatic variable individually, if we would include a time period variable it would use up a large chunk of the available variance in the model as it would in essence be a measure of a combination of all the climatic variables at once and leave little variance left for the individual climatic variables of interest.

Line 347: With only three levels for the variable of group identity, it would be better to include this as a control variable in this (and indeed all models).

We agree and now include group ID as a control variable.

Line 360: What is meant by “whole models”?

We mean the full model containing all predictor variables, interactions and random effects. This has been re-written to read “full models”.

Line 377: Sampling rates per time period per individual would be informative.

We have included more detail on mean sampling rates. We give the number of samples collected per time period in the methods. Sample rates were a maximum of 2 per individual per time period. Not all individuals appear in all time periods due to migrations and deaths of individuals.

Table 1/line 711: This table would be better in time order to illustrate the gradual increase in GCs over time, peaking in hot-dry before a slight decline. Table would then reflect the figures better.

This table has been reordered as suggested by the reviewer.

Line 379/447-448: Can you interpret single effects when the interaction is meaningful?

We agree with the reviewer that it is tricky to interpret main effects if the variable is involved in an interaction. We mention this outcome in the results section but have now made it clearer that this should be interpreted with caution as it is involved in the interaction.

Line 393: Typo for “concentrations”

This change has been made.

Line 407: Meaningful influence of time period?

This change has been made.

Line 420: “hot summer period” – better to be consistent in terminology for the different time periods if this is what you are referring to here.

This change has been made.

Line 421-429: How do you interpret temp being meaningful in one of these models and not the other?

It may be that at the monthly level food availability and water availability follow a more similar pattern. Although the VIFs indicate no multicollinearity it could be that they still follow similar patterns across months and so one predictor masks the effect of the other and the food availability variable to some extent also includes much of the water availability variable in this case. As low rainfall leads to a shortage of both food and water. Whereas temperature is a more cyclical climatic variable across the year and so in the model with both temperature and water availability the patterns of temperature and water availability would be less closely in sync. Additionally, the temperature variable was a statistical trend in one model and non-significant in the other. So the difference may not have been so great.

We have now re-run this model with the 2-day temperature variable. In doing so, the multicollinearity (indicated by high VIFs) is no longer an issue (VIFs less than 2.6 in the new model). Thus we can include temperature, food availability and water availability all in the same model.

Line 432: Phrasing is confusing, should it not be “negative influence on survivorship” or “positive influence on probability to die”?

This change has been made.

Line 475: Jarret et al not in reference list

This reference has been added to the paper.

Line 500: “lactation played a major role in the mortality patterns”.

This change has been made.

Line 531: "Here, we add..."

This change has been made.

Line 534: "... populations."

This change has been made.

Line 545: I wonder if more exploration of this result is merited. It is interesting that there is inter-individual variation in plasticity to social stressors but not climatic stressors, especially as the social time budget remains fairly constant throughout the study in spite of challenging climate. What does this say about the significance of social vs climatic environment to physiology in this species?

We have added further discussion of this result higher up in the discussion section.

Line 583: Reference lacks volume or page numbers.

The relevant information has been added.

Line 692: This Young et al appears after the other in main body text, so needs to be Young et al 2017b rather than 2017a.

We have reordered the citations appropriately.

Line 737: No bold highlighting apparent in the table

Bold typing has been added.

Figure 2: X-axis appears to be standardised by this is not stated in the figure or its legend.

This correction has been made.

Reviewer: 2

Comments to the Author(s)

Review ms RS0S-191078 Climate induced stress and mortality in vervet monkeys
by Julie Duboscq julie.duboscq@mnhn.fr

This study investigates the link between physiological and behavioural patterns of vervet monkeys and extreme environmental conditions such as drought, low food availability and high temperatures in the framework of the reactive scope model. The study is conducted on 3 different groups of monkeys living in a game reserve in South Africa. Results show that monkeys have more elevated fGCM during harsh periods of low food and water availability, that they change their behaviour accordingly, that there is a positive relationship between harsh conditions and high mortality in females and juveniles, and that higher fGCMs were associated with an increased probability of mortality. This is an important study to assess

how animals deal with extreme environmental conditions which will become more frequent in the future due to climate change. Overall, this is a well-designed study with good foundations, clear objectives and masterful analyses. The only serious issue might concern the way food availability has been assessed and considered which the authors should address clearly. More details on how things were done are also needed here and there.

Thank you for your positive and constructive comments in regards to our manuscript.

I know it is by no means the authors' fault and this is probably more of a message to the RS journals, I just want to remark that having the figures and tables + their legends not imbedded within the text makes for a difficult read hence assessment of the paper... isn't there an option to send stuff all tied together?

We have embedded the figures and tables in the revision to allow for better readability of the manuscript.

Kudos to the authors for sharing their dataset and R codes! What would be even more awesome would be 1/ (more detailed) annotated scripts and 2/ an explanatory note for each table/sheet of the dataset. For instance, in the model 2 sheet, what are Month1, Month2, Month3 referring too? I'm guessing it is the dummy coding of the period variable? Otherwise, it is not enough to fully replicate the analyses.

We have improved the annotation of our data to make it more accessible to the reader.

Line 72: How about being a bit more specific in this example by explicitly linking drought to starvation to death?

We have added more explanation to the text.

Lines 74-124: great explanations but not very concise. It might be a bit unusual in the introduction but how about adding a schematic/graph here to make those paragraphs a bit shorter and perhaps clearer?

We have reduced the amount of detail in the introduction dedicated to the reactive scope framework and its explanation as also recommended by Reviewer 1.

Line 126: how do we know already that the drought period had an impact on vervets' "physiological stress levels, activity and survival probabilities"? is there a verb missing here maybe to state that "we take advantage of ... to investigate/study/test ..."?

We have altered the text accordingly.

Line 133: what is "standing water"?

Our vervet study site is a natural environment with no artificial waterholes or provisioning. By standing water we refer to the fact that the river, which flows through the field site has dried up and as a result the pools, which formed in the deepest parts of the river when it stops flowing, have also dried up. So there is no water available in the monkey's home range. We have added respective details to the text to explain this more clearly.

Line 134: noticed here but may have to be defined earlier: what's a drought?

A drought is defined as a period of abnormally low rainfall leading to a shortage of water. We have added this to the text.

Lines 172 and § fwd: the readers of RSOS might not be familiar with the assessment of dominance rank in animals so it needs to be specified which behaviours are used, how data are collected, processed and arranged to be able to calculate DS... what are DSs? What's a winner/loser matrix? How to build a hierarchy from dominance ranks? "...standardized David's Scores" why and how to do that?

We have added more detail to our dominance rank section.

Line 197: this makes 332 samples but there seems to be 346 fGCM values. Why the discrepancy?

Thank you for spotting this typo with our values. We have corrected the discrepancy and added the correct values for each time period. All values now sum to 346.

Lines 198 and § fwd: it might be useful to specify that fecal samples are individually collected, i.e. from identified individuals observed pooping. Lines 201-202: directly where the feces were deposited? On a leaf? Ground? Piece of plastic?... line 203: "during" instead of "until"? Lines 205-212: how about fleshing out a bit more how those analyses were conducted? A short summary would be nice.

More details have been added to the text.

Lines 227-228: it is not clear how this assessment is made, e.g., if water availability is scored as "absent" does it mean the river is gone too? Is there any data on water consumption by the monkeys? Of course, if water is totally absent of their home range, they literally cannot drink but there must be a gradient of water availability influencing their drinking behavior?

We measured water availability on a daily basis and for each group separately. The home range consists of a river, which runs through the middle of the study site, and this is the only source of water for the monkeys. Occasionally during heavy rain showers water would collect in small holes on rocks but this would dry out within a day or two. Therefore, we were able to assess the water availability via whether water was available or not in the river. We recorded each day if the river was flowing or if there were pools of water available in the home-range of each group or if water was completely absent. Due to the limited locations monkeys are able to access water we are fully confident we recorded when water was available or absent. When pools of water were present in the home range we observed the monkeys drinking at these pools every day until the pools ran dry.

Line 229: why thirty days here when temperature is tested at a 5- and 10-days range?

We used 30 days for water availability, as we were trying to capture the cumulative effect of a chronic shortage of water (given that this is what constitutes a drought – i.e., below-average precipitation in a given region, resulting in prolonged shortages in the water supply, whether atmospheric, surface water or ground water). Using a shorter window wouldn't capture

chronic shortage, and potentially would generate a false impression of how much water was present in the home range. That is, during the drought period, the river dried up completely, with the result that there was no access to water for the monkeys for several days and even weeks at a time. When it did rain, high temperatures (in excess of 45°C) meant that any small pools of water dried up rapidly. So, if a faecal sample was collected in the one or two days following a rain shower, but this was actually the only rainfall for the whole month and water was present for only a very few days, it would give a misleading impression of how much water was available to the monkeys overall.

Line 232 and § fwd: I get that this is all research-in-progress and that a single project results in many publications with more or less related or similar (or exact same) methodologies but for a 1st time reader, it is hard to read that x,y,z methods was used, read papers x,y,z to understand how it is done. It is ok to do so if a summary of the method is given then other publications are referenced for MORE details (emphasis on more, some details are necessary first). As for endocrinological analyses lines 205-212, it would be helpful to have some details here too for instance how is the amount of biomass measured (colour gradient? Luminosity? Light reflection?...)? what does “more abundant in biomass” mean in terms of food availability and how and why is this really a good proxy? The vervets are not folivorous are they? (How) Does foliage correlate with fruiting? Was there any test of this relationship? Maybe this all seems fairly obvious but it would be better to spit it out.

We have added additional details in our methodology section to make them all more comprehensible to a wider audience. Vervet monkeys eat a wide variety of plant matter from seeds and small fruits to leaves. We do not experience fruiting seasons of large fruiting trees such as figs as one may see in the rainforest. The study by Willems et al (2008) looked at phenology data collected on the ground and compared it to the satellite imagery and the NDVI and found a strong correlation between the two methods for vervet monkeys in Southern Africa. Therefore, the NDVI measurement has been shown in similar habitats to be a good proxy for the food available for the vervet monkey population. Additionally, we have shown a similar pattern for the study population more recently (Dostie et al 2019).

Lines 259-261: what was the rationale behind those numbers, apart from it is used/proposed as settings in the papers cited, in short?

These numbers represent:

- 1) Cores: the number of core processors used for the analysis, the computer used for the analysis determines this.*
- 2) Iterations: the number of posterior samples taken during the analysis in order to determine the result. If a model is unstable and does not converge then the number of iterations can be increased to increase the number of samples taken and help convergence. Posterior checks of the sample distribution and \hat{r} indicate if the model converges.*
- 3) Priors: As we do not have precise previous knowledge of the effect of these variables on our response variable we used weakly informative priors, which is recommended in such circumstances.*

Line 269: individual fGCM concentration per fecal sample? Per 6-weeks periods? In the xls file, there's between 2 and 8 values per individual monkey, what do they represent? Wouldn't it be important to control for either number of samples per individuals or date of collection or is that implicit in controlling for ID and testing for environmental variables?

We control for ID as a random effect and for each of the six-week periods we have distinct variation in our climatic variables. For each time period, we have only two samples per individual but due to deaths, emigrations or migrations we do not have every individual in every one of the four time periods. This explains the difference in sampling number. We do not include time periods as a variable as due to the distinct differences in climatic variables of these four time periods. The model is designed to test the differences of climatic variables between these four time periods but on each climatic variable individually, if we included a time period variable it would use up a large chunk of the available variance in the model as it would in essence be a measure of a combination of all the climatic variables at once and leave little variance left for the individual climatic variables of interest.

Line 270 2): the NDVI index then?

More detail has been added.

Lines 273-277: what was the rationale behind the choice of interactions? Why is dominance rank used as a control factor, to control what exactly? And what are random slopes and intercepts for?

We chose the interaction between food availability or temperature with water availability as the latter is predicted to amplify the effect of the other two variables on the physiology of the animal, i.e., at extreme temperatures or with low food availability, the effects of these variables should be amplified by the absence of water. To examine this we included the interaction.

Additionally, we included rank as many previous studies have shown rank effects on resource competition (access to food, water or preferred resting spots to cope with adverse conditions), so we wanted to control for this in our models as our specific interest here is the impact of climatic variables on fGCM levels. We did not include additional social factors, and this we mention in the discussion.

We included random slopes and intercepts in order to examine individual level effects, i.e., to examine whether any of our predictor variables acted at the individual rather than/in addition to the population level.

Line 289: how about a short explanation of this criterion? E.g. (how) Is it related to the other(s) AIC? Why this one?

The WAIC is the widely applicable information criterion, which is the generalised version of the AIC. It acts in the same manner as the AIC score but can be used across a wider range of models such as Bayesian models. The lower the WAIC score the better fit to the model as with the AIC.

Lines 298 and § fwd: isn't it necessary to imbed behaviours into their scans? I.e. individual A moved in scan 1, rested in scan 2, moved again in scan 3 ect ect. Because there is a full list of behaviours for all individuals in each scan, right?

This multinomial model estimated the relative probability of the 4 behaviours occurring during a scan. We used this model to estimate how these relative probabilities changed depending on which individual was observed, it's fGCM level, sex, rank, and the time period of

the observation. We therefore only estimated average probabilities. It would be, however, interesting to test a more sequential approach (A moved, then rested, then ate... etc), but this would be beyond the scope of this paper.

Line 305: why was moving considered the reference category, and not e.g. the most frequent activity? If I understood correctly how the data are organized here, there are, let's say, 100 scans for individual A in period 1 and 115 in period 2, so there are 100 and 115 behaviour values respectively but 100 and 115 times the same fGCM value, 100 and 115 times the same standardized rank, ect. How does the model deal with this and is it an issue?

As the multinomial is only estimating the probability of each behaviour the fact that the absolute number of scans varies is not an issue. Though if the number of scans is very low this can be an issue. This is why we used a multilevel multinomial model to help deal with unbalanced data (i.e., some have more scans than others). The reference category could have been anyone of the four behaviours. The model estimates the probability of each behaviour against the other three behaviours and gives a probability of the four occurring. It was a random selection of resting as our reference but using any of the four behaviours would give the same output.

Lines 340 and § fwd: reading the text and looking at the Model4 datasheet, I don't manage to make sense of how this works. What are T2 and T1 in the datasheet?

The model is a time series cox proportional hazards model and T1 and T2 are the start and end times for each period. These are basically reference categories used by the model to determine if the individual was present or not, and to provide the sequence of events. We have added more detail to the methods to make this clearer.

Line 352: what was the issue with this influential case?

When we ran the DF beta tests, one data point was flagged as an outlier pulling the results in one direction, as indicated by an extremely large DF beta value for the days without water variable. Therefore, this data point was skewing the data heavily in one direction. As a result we removed this value and re-ran the model with all DF beta values within the recommended range.

Lines 379, 380, ect: "meaningful" effect or interaction is another way of saying significant without saying significant? It can be meaningful without being significant and the other way around so use another term, or no qualifier at all, or stick with significant.

Our analysis is a Bayesian analysis and so to use the word significant is not appropriate, as we are not using null-hypothesis testing in our approach. And so as a result we are not assessing the significance of our findings but we are updating our hypotheses in the light of our data. A posterior estimate can be meaningful but it cannot be "significant" in the statistical sense because we are not generating p-values in that way.

Lines 380-381: it would be better to describe the effect of the interaction first then of the single variable.

This change has been made.

Lines 407 and § fwd: I don't understand the influence of time here? How does the model tell that variable in period x is higher/lower than variable in period y? Wouldn't this all be within normal variation?

What the model is saying here is that the probability of seeing one of the four behaviours during a scan varied by time period. Here we used the 95%CI to help identify changes in these probabilities.

Lines 424-425 and 428-429: How can this be? Maybe a sentence or 2 of interpretation would already make sense here?

It may be that at the monthly level food availability and water availability follow a more similar pattern. Although the VIFs indicate no multicollinearity it could be that they still follow similar patterns across months and so one predictor masks the effect of the other and the food availability variable to some extent also includes much of the water availability variable in this case. As low rainfall leads to a shortage of both food and water. Whereas temperature is a more cyclical climatic variable across the year and so in the model with both temperature and water availability the patterns of temperature and water availability would be less closely in sync. Additionally, the temperature variable was a statistical trend in one model and non-significant in the other. So the difference may not have been so great.

We have now re-run this model with the 2-day temperature variable. In doing so the multicollinearity (indicated by high VIFs) is no longer an issue (VIFs less than 2.6 in the new model). Thus we can include temperature, food availability and water availability all in the same model.

Line 457: the or these animals instead of our animals

This change has been made.

Line 478 and § above: or the timescale of analysis is not fine enough to capture individual differences?

This could be true and we have added such a point to the discussion.

Line 503 and § above: so, maybe it would have been judicious to include other individual factors, such as "reproductively active" i.e. engaged in sexual activities vs "non-active", "pregnant" vs "nonpregnant", "lactating" vs "nonlactating"?

Due to the seasonality in vervet monkey reproductive states we decided against including individual reproductive states using the same rationale as to why we did not include time period in our analysis. Our four selected time periods would fall into two categories of the mating season (cold-dry and cold-wet) and just after the birth season when females will be lactating (hot-dry and hot-wet). Thus due to the seasonality of the reproductive states and the fact that the vast majority of our females gave birth each year, we think that adding such a

factor would overlap greatly with our test predictors. However, we agree that a more fine scaled longer term analysis including reproductive state in it would be of great interest.

Line 544: it is weird to mention this just now and here and not discuss it anywhere else, like § starting line 468 maybe or line 504.

We have added further discussion of this point higher in the discussion section.

Table 1: I would also add the dates of those periods in the first column, just as a reference.

This column has been added.

Table 2 (and elsewhere): “certainty” intervals? The text says “credible” intervals line 255. Are they one and the same? Then consistency in term use is warranted I guess.

We have changed all certainty intervals to credible intervals to provide consistency of the terms.

Table 3: can the periods be labeled consistently here too? either with a date or with a qualifier like in table 1 or both but not one or the other.

This change has been made.

Figure 1: it would also be helpful to highlight in this graph that the purple fecal sample collection periods are also the analysed periods, i.e. “cold-wet” (April-June 2015), “hot-wet” (December 2015-February 2016), “hot-dry” (December 2016-February 2017) and “cold-dry” (April-June 2017). Furthermore what NVDI values are used here? A monthly average from the bi-monthly downloaded images? The value “relating to the closest image prior to fecal sample collection” (line 242)? Indicate again sample size and which model it does refer to.

This graph is a schematic graph to illustrate how the climatic variables vary across the study period and to give an overview of how they differ over time and at what time point data was collected. It does not refer to any of the statistical models directly. We have added additional information to the figure legend as suggested by the reviewer.

Figure 2: why 24 out of 30? There’s a typo in food availability. Indicate again sample size and which model it does refer to.

The graph is determined from the marginal effects plots. With an interaction the marginal effects plot sets one variable (in this case water availability) to three levels and plots it against the other variable (in this case food availability). The two extremes and the mean water availability were selected, so water available for no days, all the days and for 24 out of 30 days, which was the mean value from the dataset.

Figure 3: it would be helpful to specify in the legend what a probability of 01 or 06 means (y-axis) and what the grey area means too. Indicate again sample size and which model it does refer to.

We have added additional text to the legend to make this clearer to the reader.

Figure 4: x-axis title says “week of study” but legend says “periods” which if I understood correctly last 2 weeks bc bi-monthly frequency, right? So it’s a bit incorrect. Indicate again sample size and which model it does refer to.

We agree that this x-axis labelling can be confusing. We have changed the label to “fortnightly period of study”. The model name has been added to the legend.

Figure 5: what’s the threshold for the split? Indicate again which individual data make that plot, i.e. females and juveniles, and which model it does refer to.

This graph is for illustrative purposes; the actual variable used fGCM concentrations as a continuous variable. Here we split the data at the mean, with those above the mean considered to show high fGCM levels and those below the mean low levels. The graph refers to Model4_{Surv+food} and includes all adult individuals, as these are the individuals we have fGCM levels for. Due to the change in the temperature variable this figure and the model have been re-run and changed appropriately.

Appendix B

Christopher Young Ph.D
Mammal Research Institute,
University of Pretoria
Faculty of Natural and Agricultural Science,
Pretoria,
Republic of South Africa
Christopher.young@uleth.ca

Ref: RSOS-191078.R1

Pretoria, 6th October 2019

Dear Dr. Carter, Mr. Dunn,

We would like to thank you for the acceptance of our paper for publication with minor revisions in the Royal Society Open Science (RSOS-191078.R1: *Climate induced stress and mortality in vervet monkeys*). We are very happy with the decision and have made the suggested changes to the manuscript.

Please find below our detailed response to the Associate Editors comments (marked in red and italics). Thank you again for your acceptance of our manuscript.

Yours sincerely,

Dr. Christopher Young (on behalf of my co-authors)

Associate Editor Comments to Author (Dr Alecia Carter):

Associate Editor

Comments to the Author:

Dear authors,

Thank you for taking the time to carefully address the reviewers' comments on your earlier submission. I have only a few small grammatical corrections to suggest, outlined below.

Thank you for submitting this manuscript to RSOS.

We thank the Associate Editor for their acceptance of our manuscript and have made all of the suggested changes listed below.

L74. Please move the Latin species name to after the mention of the species.

L120. monkeys -> monkeys'

L211 and elsewhere: space required between number and unit

L222: weighted -> weighed

L253: please split this paragraph here

L313: a semi-colon would be more appropriate here (after the reference) than a comma

L547: more likely -> greater (as you already mention likelihood earlier in the sentence)

L626: morality -> mortality (though I do like the implied idea of the goodness of the mothers' ultimate sacrifices)

L636: animal's -> animals'

L653: in wild -> in a wild